



# A Comparative Study of Two-way and Offline Coupled WRF v3.4 and CMAQ v5.0.2 over the Contiguous U.S.: Performance Evaluation and Impacts of Chemistry-Meteorology Feedbacks on Air Quality

Kai Wang[1], Yang Zhang[1*], Shaocai Yu[2*], David C. Wong[3], Jonathan Pleim[3], Rohit Mathur[3], James T. Kelly[4], and Michelle Bell[5]

[1]Department of Civil and Environmental Engineering, Northeastern University, Boston, MA 02115
[2]Key Laboratory of Environmental Remediation and Ecological Health, Ministry of Education; Research Center for Air Pollution and Health, College of Environment and Resource Sciences, Zhejiang University, Hangzhou, Zhejiang 310058, P.R. China
[3]Center for Environmental Measurement and Modeling, U.S. EPA, RTP, NC 27711
[4]Office of Air Quality Planning and Standards, U.S. EPA, RTP, NC 27711
[5]School of Forestry & Environmental Studies, Yale University, New Haven, CT 06511

*Correspondence to: Yang Zhang (ya.zhang@northeastern.edu); Shaocai Yu (shaocaiyu@zju.edu.cn)





## Abstract

The two-way coupled Weather Research and Forecasting and Community Multiscale Air
Quality (WRF-CMAQ) model has been developed to more realistically represent the atmosphere
by accounting for complex chemistry-meteorology feedbacks. In this study, we present a
comparative analysis of two-way (with consideration of both aerosol direct and indirect effects)
and offline coupled WRF v3.4 and CMAQ v5.0.2 over the contiguous U.S. Long-term (five-year
of 2008-2012) simulations using WRF-CMAQ with both offline and two-way coupling modes
are carried out with anthropogenic emissions based on multiple years of the U.S. National
Emission Inventory and chemical initial and boundary conditions derived from an advanced
Earth system model (i.e., a modified version of the Community Earth System Model/Community
Atmospheric Model). The comprehensive model evaluations show that both two-way WRF-
CMAQ and WRF-only simulations perform well for major meteorological variables such as
temperature at 2 m, relative humidity at 2 m, wind speed at 10 m, and precipitation (except for
against the National Climatic Data Center data) as well as shortwave/longwave radiation. Both
two-way and offline CMAQ also show good performance for ozone ($O_3$) and fine particulate
matter ($PM_{2.5}$). Due to the consideration of aerosol direct and indirect effects, two-way WRF-
CMAQ shows improved performance over offline-coupled WRF and CMAQ in terms of
spatiotemporal distributions and statistics, especially for radiation, cloud forcing, $O_3$, sulfate,
nitrate, ammonium, and elemental carbon as well as tropospheric $O_3$ residual and column
nitrogen dioxide ($NO_2$). For example, the mean biases have been reduced by more than 10 W m$^{-2}$
for shortwave radiation and cloud radiative forcing and by more than 2 ppb for max 8-h $O_3$.
However, relatively large biases still exist for cloud predictions, some $PM_{2.5}$ species, and $PM_{10}$,
which warrant follow-up studies to better understand those issues. The impacts of chemistry-





meteorological feedbacks are found to play important roles in affecting regional air quality in the
U.S. by reducing domain-average concentrations of carbon monoxide (CO), $O_3$, nitrogen oxide
($NO_x$), volatile organic compounds (VOCs), and $PM_{2.5}$ by 3.1% (up to 27.8%), 4.2% (up to
16.2%), 6.6% (up to 50.9%), 5.8% (up to 46.6%), and 8.6% (up to 49.1%), respectively, mainly
due to reduced radiation, temperature, and wind speed. The overall performance of the two-way
coupled WRF-CMAQ model achieved in this work is generally good or satisfactory and the
improved performance for two-way coupled WRF-CMAQ should be considered along with other
factors in developing future model applications to inform policy making.
**Keywords:** CMAQ, Two-way coupling, Evaluation, Chemistry-meteorology feedback
**1. Introduction**

The Community Multiscale Air Quality (CMAQ) modeling system developed by the U.S.

Environmental Protection Agency (EPA) (Byun and Schere, 2006; Scheffe et al., 2016; San
Joaquin Valley APCD, 2018; Pye et al., 2020; U.S. EPA, 2020) has been extensively used by
both scientific community and governmental agencies over various geographical regions and
under different meteorological and air pollution conditions to address major key air quality
issues such as atmospheric ozone ($O_3$), acid rain, regional haze, and trans-boundary or long-
range transport of air pollutants during the past decades over North America (Zhang et al.,
2009a,b; Wang and Zhang, 2012; Hogrefe et al., 2015), Asia (Wang et al., 2009, 2012; Liu et al.,
2010; Zheng et al., 2015; Li et al., 2017; Xing et al., 2017; Yu et al., 2018; Mehmood et al.,
2020), and Europe (Kukkonen et al., 2012; Mathur et al., 2017; Solazzo et al., 2017). The
CMAQ model is traditionally driven offline by the three-dimensional meteorology fields
generated separately from other meteorological models such as the Weather Research and
Forecasting (WRF) model, and the dynamic feedbacks of chemistry predictions on meteorology



are neglected. However, more recently (IPCC, 2018), chemistry-meteorology feedbacks have
been found to play important roles in affecting the both global and regional climate change and
air quality (Jacobson et al., 1996; Mathur et al., 1998; Ghan et al., 2001; Zhang, 2008; Zhang et
al., 2010, 2015a,b, 2017; Grell and Baklanov, 2011; Wong et al., 2012; Baklanov et al., 2014; Yu
et al., 2014; Gan et al., 2015a; Wang et al., 2015; Xing et al., 2015a,b; Yahya et al., 2015a,b;
Hong et al., 2017; Jung et al., 2019). Feedbacks of aerosols on radiative transfer through aerosol-
radiation interactions (i.e., aerosol direct forcing) and aerosol-cloud interactions (i.e., aerosol
indirect forcing) are especially important (Zhang, 2008; Zhang et al,, 2015a,b; Baklanov et al.,
2014; Wang et al., 2015; Yahya et al., 2015a,b). Recognizing this importance, as well as the
recent advances in knowledge on chemistry-meteorology interactions and computational
resources, the U.S. EPA developed a two-way coupled WRF-CMAQ model that accounts for the
aerosol direct effect alone (Wong et al., 2012). This version of CMAQ has been applied for both
regional and hemispheric studies (Wang et al., 2014; Hogrefe et al., 2015; Xing et al., 2016,
2017; Hong et al., 2017, 2020; Sekiguchi et al., 2018; Yoo et al., 2019). For example, Xing et al.
(2016) showed that aerosol direct feedbacks may further improve air quality resulting from
emission controls in the U.S. and also indicated that coupled models are key tools for quantifying
such feedbacks. Reduction in atmospheric ventilation resulting from aerosol induced surface
cooling can exacerbate ground level air pollution. Hong et al. (2017) estimated an increase by
4.8%-9.5% in concentrations of major air pollutants over China in winter due to incorporation of
such effects. Xing et al. (2017) reported that the aerosol direct effects could reduce daily max 1h
$O_3$ by up to 39 $\mu g\ m^{-3}$ over China in January through reducing solar radiation and photolysis
rates. Hong et al. (2020) found that the benefits of reduced pollutant emissions through
weakening aerosol direct effects can largely offset the additional deaths caused by the warming





effect of greenhouse gases over China. Some of those studies have also found that the missing
aerosol indirect effects in WRF-CMAQ may introduce large model biases on their simulations of
radiation and thus air quality (Wang et al., 2014; Sekiguchi et al., 2018; Yoo et al., 2019). There
has been a growing awareness that both aerosol effects should be considered together to provide
greater fidelity in coupling complex atmospheric processes among chemistry, aerosols, cloud,
radiation, and precipitation (Grell and Baklanov, 2011). To address this issue and better represent
the one-atmosphere modeling capability of CMAQ, Yu et al. (2014) further extended the two-
way coupled WRF-CMAQ model by including aerosol indirect effects and improved WRF-
CMAQ's capability for predicting cloud and radiation variables.

Different from the traditional online integrated air quality models such as the Gas,

Aerosol, Transport, Radiation, General Circulation, and Mesoscale Meteorological (GATOR-
GCMM) model (Jacobson, 2001), the WRF model coupled with chemistry (WRF/Chem; Grell et
al., 2005) and the WRF model coupled with the Community Atmosphere Model version 5
(WRF-CAM5; Ma et al., 2013; Zhang et al., 2015a,b; 2017), in which atmospheric dynamics and
chemistry are integrated and simulated altogether without an interface between meteorology and
atmospheric chemistry (Zhang et al., 2013), two-way WRF-CMAQ (also referred to as the online
access model) is created by combining existing meteorology (i.e., WRF) and atmospheric
chemistry (i.e., CMAQ) models with an interactive interface (Yu et al., 2014). As pointed out by
Yu et al. (2014), the main advantage of two-way CMAQ is to allow the existing numerical
techniques to be used in both WRF and CMAQ to facilitate future independent development of
both models while also maintaining CMAQ as a stand-alone model (the offline capability). In the
past, a number of studies have compared and evaluated online vs. offline-coupled model
performance (Pleim et al, 2008; Matsui et al., 2009; Wilczak et al., 2009; Lin et al., 2010;





Herwehe et al., 2011; Yu et al., 2011; Wong et al., 2012; Zhang et al., 2013, 2016a; Choi et al.,
2019). However due to the missing offline-coupled mode or component for most online-coupled
models, many of those intercomparison studies are subject to some key limitations such as
inconsistent model treatments in chemical options (Matsui et al., 2009; Lin et al., 2010; Zhang et
al., 2013; Choi et al., 2019) or in both physical and chemical options (Wilczak et al., 2009;
Herwehe et al., 2011; Zhang et al., 2016a), different domain projection methods or resolutions
(Wilczak et al., 2009; Lin et al., 2010; Zhang et al., 2013), or disunified model inputs (Wilczak et
al., 2009; Lin et al., 2010; Zhang et al., 2013). Due to the unique coupling approach, two-way
WRF-CMAQ can be used to overcome those limitations and set up ideal intercomparisons
between online and offline simulations using consistent model treatments (Pleim et al, 2008; Yu
et al., 2011; Wong et al., 2012).

In this study, we provide a robust examination of model improvements by considering

chemistry-meteorology feedbacks and their impacts on the U.S. air quality using the two-way
WRF-CMAQ model (same version as in Yu et al., 2014) with both aerosol direct and indirect
effects. Long-term (five-year of 2008-2012) simulations using both two-way and offline coupled
WRF and CMAQ models are carried out and compared to the best of our knowledge for the first
time over the contiguous U.S. (CONUS) with anthropogenic emissions based on multiple years
of the U.S. National Emission Inventory (NEI) and chemical initial and boundary conditions
(ICONs/BCONs) downscaled from the advanced Earth system model, i.e., an updated version of
the Community Earth System Model/CAM5 (CESM/CAM5; He and Zhang, 2014; Glotfelty et
al., 2017). Our objectives include 1) perform a comprehensive model evaluation for major
meteorological variables and chemical species from this long-term application of the two-way





coupled WRF-CMAQ; and 2) conduct a comparative study of two-way and offline coupled WRF
and CMAQ to examine the impacts of chemistry-meteorology interactions on U.S. air quality.

Compared to previous studies in the literature, there are a few key features of this work.

First, the intercomparisons between two-way (or online) and offline WRF-CMAQ are performed
here using consistent model configurations including both physical/chemical options and inputs.
Second, unlike a few previous intercomparison studies (Pleim et al, 2008; Yu et al., 2011; Wong
et al., 2012) using two-way WRF-CMAQ with only aerosol direct effects for relatively short
episodes, the model version in this work includes both aerosol direct and indirect effects and
simulations are conducted for multiple years to provide more robust assessments. Third,
compared to other studies (e.g., Yahya et al., 2015a,b; Choi et al., 2019) focusing on the impacts
of chemistry-meteorology feedbacks on meteorology only or limited chemical species, this study
performs comprehensive and extensive evaluation and comparison to demonstrate importance of
chemistry-meteorology feedbacks on regional meteorology and air quality.
**2. Model description, simulation setup, and evaluation protocols**

Two sets of five-year (i.e., 2008-2012) long-term simulations are conducted using the two-

way coupled WRF v3.4-CMAQ v5.0.2 model with both aerosol direct and indirect effects and
the sequentially offline-coupled WRF v3.4 and CMAQ v5.0.2 model, respectively, over the
CONUS with 36-km horizontal grid spacing. The vertical resolution for these simulations
consists of 34 layers from the surface (~38 m) to 100 hPa (~15 km). The two-way coupled WRF-
CMAQ includes estimations of aerosol optical properties based on prognostic aerosol size
distributions and composition . These aerosol optical properties are then used to modulate the
shortwave radiation budget estimated using the Rapid and accurate Radiative Transfer Model for





General circulation (RRTMG) radiation scheme (Iacono et al., 2008) in WRF. Additionally,
aerosol indirect effects, including the first (cloud albedo) and second (cloud lifetime) indirect
aerosol forcing and the glaciation (ice and mixed-phase cloud lifetime) indirect aerosol forcing
are also modeled. More details on the model development of this version of WRF-CMAQ can be
found in Yu et al. (2014). On the other hand, the WRF only model calculates the radiation
budgets by using prescribed aerosol optical properties such as aerosol optical depth, single
scattering albedo and asymmetry parameters and cloud formation by assuming default droplet
number concentration and fixed cloud effective radius, which may not be representative for the
large regions with complex air pollution conditions. Both the two-way and offline coupled WRF-
CMAQ use the same model configurations as shown in Table S1 in the supplementary material,
except that prognostic aerosol impacts on radiation and clouds are fully treated in two-way
WRF-CMAQ. The physics options include the RRTMG shortwave and longwave radiation
schemes, the Asymmetric Convective Model (ACM2) planetary boundary layer (PBL) scheme
(Pleim, 2007), the Pleim-Xiu (PX) land-surface scheme (Xiu and Pleim, 2001), the Morrison
two-moment microphysics scheme (Morrison et al., 2009), and version 2 of the Kain–Fritsch
(KF2) cumulus scheme (Kain, 2004). The chemical options include the Carbon Bond 2005
(CB05) chemical mechanism (Yarwood et al., 2005) with additional chloride chemistry (Sarwar
et al., 2008), the sixth generation CMAQ aerosol module (AERO6) (Appel et al., 2013), and
CMAQ's aqueous phase chemistry (AQCHEM). In addition, the time steps of dynamics and
radiation for two-way WRF-CMAQ are set as 1 min and 15 mins, respectively, and the call
frequency for CMAQ in the two-way coupled model is set to be 5 mins.

The meteorological ICONs/BCONs are generated from the National Centers for

Environmental Prediction Final Analysis (NCEP-FNL) datasets and the chemical



ICONs/BCONs are downscaled from a modified version of CESMv1.2.2/CAM5 (He and Zhang,
2014; Glotfelty et al., 2017). The anthropogenic emissions are based on two versions of NEI.
NEI 2008 and NEI 2011 are used to cover the 5-year period, i.e., NEI 2008 for 2008-2010 and
NEI 2011 for 2011-2012, respectively. Biogenic emissions are calculated online using the
Biogenic Emissions Inventory System (BEIS) v3 (Schwede et al., 2005). The sea-salt and dust
emissions are also generated online by CMAQ's inline modules (Zhang et al., 2005; Foroutan et
al., 2017). Two-way coupled WRF-CMAQ simulations are reinitialized every 5 days to make
meteorology simulations as accurate as possible while preserving the two-way chemistry-
meteorology feedbacks. The WRF-only simulations that are used to drive the offline CMAQ
simulations apply the same reinitialization method to be consistent with the two-way coupled
WRF-CMAQ simulations.

The model evaluation in this work mainly focuses on the long-term climatological type of

performance by comparing 5-year average spatially and temporally matched model predictions
of major surface meteorological/radiation-cloud variables and surface/column chemical species
against various surface/satellite observations and reanalysis data. The surface meteorological
data include temperature at 2 m (T2), relative humidity at 2 m (RH2), wind speed at 10 m
(WS10), and wind direction at 10 m (WD10) from the National Climatic Data Center (NCDC),
and precipitation from the NCDC, the National Acid Deposition Program (NADP), the Global
Precipitation Climatology Project (GPCP), the Parameter-elevation Regressions on Independent
Slopes Model (PRISM), and the Tropical Rainfall Measuring Mission Multisatellite Precipitation
Analysis (TMPA). The radiation and cloud data include downward shortwave radiation at the
ground surface (SWDOWN), net shortwave radiation at the ground surface (GSW), downward
longwave radiation at the ground surface (GLW), outgoing longwave radiation at the top of the



atmosphere (OLR), and shortwave and longwave cloud forcing (SWCF and LWCF) from the
Clouds and the Earth's Radiant Energy System (CERES); aerosol optical depth (AOD), cloud
fraction (CF), cloud water path (CWP), and cloud optical thickness (COT) from the MODerate
resolution Imaging Spectroradiometer (MODIS); and cloud droplet number concentration
(CDNC) derived based on MODIS data by Bennartz (2007). The chemical data include surface
$O_3$ from the Aerometric Information Retrieval System-Air Quality Subsystem (AIRS-AQS) and
the Clean Air Status and Trends Network (CASTNET); surface fine particulate matter ($PM_{2.5}$)
and its constituents including sulfate ($SO_4^{2-}$), nitrate ($NO_3^-$), ammonium ($NH_4^+$), elemental
carbon (EC), organic carbon (OC), and total carbon (TC = EC + OC) from the Interagency
Monitoring of Protected Visual Environments (IMPROVE) and the Chemical Speciation
Network (CSN); surface coarse particulate matter ($PM_{10}$) from the AQS; and column abundance
variables such as column carbon monoxide (CO) from the Measurements of Pollution in the
Troposphere (MOPITT), tropospheric ozone residual (TOR) from the Ozone Monitoring
Instrument (OMI), and column nitrogen dioxide ($NO_2$) and formaldehyde (HCHO) from the
Scanning Imaging Absorption Spectrometer for Atmospheric Chartography (SCIAMACHY).

The satellite datasets used in this study are all level-3 gridded monthly-averaged data

with various resolutions (i.e., 0.25° for OMI and PRISM, 0.5° for SCIAMACHY, 1° for CERES,
GPCP, MODIS, and MOPITT). For the calculation of model performance statistics, the satellite
data with different resolutions are mapped to CMAQ's Lambert conformal conic projection
using bi-linear interpolation in the NCAR command language. CMAQ model outputs at
approximate time of the satellite overpass are paired with the satellite retrievals to facilitate a
consistent comparison. Modeled CDNC is calculated as the average value of the layer of low-
level warm clouds between 950 and 850 hPa as suggested by Bennartz (2007). Following the



approach of Wielicki et al. (1996), the SWCF and LWCF are calculated as the difference
between the clear-sky and the all-sky reflected radiation at the top of atmosphere for both
simulations and observations.
The statistical performance evaluation follows a protocol similar to that of Zhang et al.
(2006, 2009a) and Yahya et al. (2016) and uses well-accepted statistical measures such as
correlation coefficient (R), mean bias (MB), root mean square error (RMSE), normalized mean
biases (NMB), and normalized mean error (NME) (S. Yu et al., 2006). Because of different
sampling protocols among monitoring networks, the evaluation is conducted separately for
individual networks for the same simulated variables/species.
**3. Comprehensive model evaluation of two-way WRF-CMAQ**
**3.1 Meteorological evaluation**
**3.1.1 Surface meteorological variables**
Figures 1a-d show the spatial distribution of 5-year average MBs for T2, RH2, WS10,
and hourly precipitation from two-way WRF-CMAQ against the NCDC data in 2008-2012 and
Table 1 summarizes the statistics for the same variables. All variables except for precipitation
show overall good or moderate spatial performance with many sites showing MBs within ±0.6
˚C for T2, ±5 % for RH2, ±1 m s$^{-1}$ for WS10, and ±0.1 mm hr$^{-1}$ for precipitation, respectively.
WRF-CMAQ tends to overpredict T2 (i.e., warm bias) over widespread areas of domain
especially along the Atlantic coast, the eastern/southeastern U.S., the Central U.S., and Pacific
coast. The model also shows cold biases (i.e., underprediction in T2) over the mountainous
regions and northeastern U.S. Similar warm biases of T2 have been previously reported by
Cohen et al. (2015) and are found to be associated with the relatively deeper PBL depth using the





non-local ACM2 PBL scheme. The relatively larger warm/cold biases over coastal and
mountainous areas are likely caused by the coarse spatial grid spacing of 36-km which cannot
resolve the complex topography. Compared to many previous WRF studies (Wang et al., 2012;
Brunner et al., 2015; Yahya et al., 2016), which typically show cold T2 biases, the overall small
warm biases in this study can be attributed to the soil moisture nudging technique used in the PX
land surface scheme (Pleim and Gilliam, 2009). The spatial patterns of MBs for RH2 show a
clear anti-correlation compared to T2 (i.e., RH2 is overpredicted where T2 is underpredicted and
vice versa). This is consistent with how RH2 is calculated based on T2. The spatial distribution
of MBs for WS10 also shows dominant overpredictions especially along coastlines, indicating
the prescribed sea-surface temperature might not be sufficient to resolve the air-sea interactions.
Systematic overpredictions of hourly precipitation against NCDC data are found to be mainly
caused low non-convective precipitation events and should be attributed to the uncertainties
associated with the Morrison microphysics scheme (Yahya et al., 2016).

The precipitation performance is further examined by comparing WRF-CMAQ with

GPCP and PRISM as shown in Figures 1e-g. The spatial distribution of precipitation is well
simulated by WRF-CMAQ especially over the land against both GPCP and PRISM by capturing
the hot spots along the Pacific Northwest coast and some areas over eastern U.S. Moderate
overpredictions of precipitation against GPCP over the Atlantic Ocean and Gulf of Mexico are
also evident, possibly due to overprediction of convective precipitation intensity by the Kain–
Fritsch cumulus scheme (Hong et al., 2017) over ocean. As shown in Table 1, the domain-
average statistics demonstrate good performance for all variables except for precipitation against
NCDC in terms of MBs, NMBs, RMSE, and Rs. For example, the MBs for T2, RH2, WS10, and
precipitation are 0.1 °C, 2.2%, 0.44 m s$^{-1}$, and 0.14-0.28 mm day$^{-1}$, respectively, and Rs for those



variables are typically between 0.5-0.98, which are well within the performance benchmark
values recommended by Zhang et al. (2013) and Emery et al. (2017).

**3.1.2 Radiation and cloud variables**

Figure 2 compares the 5-year average spatial distribution of major radiation variables
(i.e., SWDOWN, GSW, GLW, OLR, and AOD) based on the satellite retrievals and two-way
WRF-CMAQ simulations, and Table 1 summarizes the domain-average model performance
statistics. WRF-CMAQ predicts the longwave radiation variables GLW and OLR very well with
domain-average of NMBs of -1.9% and 0.8%, respectively, and Rs of 0.99 for both. The
shortwave radiation variables SWDOWN and GSW are overpredicted on average with NMBs of
13.0% and 11.1%, respectively, and Rs of 0.97 for both. The simulations also reliably reproduce
the spatial distribution of both longwave and shortwave radiation compared to observations. The
relatively large overpredictions for shortwave radiation are very likely caused by the
underpredictions of aerosol direct radiative forcing reflected from the underpredictions of AOD
(Figure 2) as well as underprediction of indirect cloud radiative forcing (see Figure 3). It has
been reported that WRF v3.4 does not treat the subgrid cloud feedback to radiation, which could
also contribute to the overpredictions in shortwave radiation (Alapaty et al., 2012; Hong et al.,
2017). The model largely underpredicts the magnitude of AOD (NMB: -64.8%), while providing
a reasonable representation of the spatial distribution of AOD over the U.S., with generally
higher values in the east and lower values in the west. The model also underpredicts the elevated
AODs over oceans and the northern part of domain. Similar AOD underpredictions have been
reported in previous studies over the U.S. using two-way coupled WRF-CMAQ (Gan et al.,
2015a; Hogrefe et al., 2015; Xing et al., 2015a). The relatively large underpredictions of AOD
may be caused by several factors. First, underprediction of $PM_{2.5}$ concentrations, particularly



$SO_4^{2-}$ and OC (Table 2), can contribute significantly to the underprediction of AOD, especially
over the eastern U.S.  Second, the underestimation of dust emissions may contribute to missing
hot spots from the model over arid areas in CA and AZ (Foroutan et al., 2017) and
underestimates of sea-salt emissions may lead to missing elevated AODs over oceans (Gan et al.,
2015b). Third, challenges in adequately representing prescribed and wildfire emissions in the
NEI (Kelly et al., 2019) may cause many missing hot spots over large areas of the Pacific
Northwest, CA, Canada, and the eastern U.S. Fourth, uncertainties in BCONs of $PM_{2.5}$
concentrations may further contribute to underpredictions of AOD over oceans and the northern
part of the domain. For example, Kaufman et al. (2001) found that the background AOD could
reach 0.1 over the Pacific Northwest using Aerosol Robotic Network (AERONET) data. The
AODs in the current simulation seem to be biased low (between 0.06-0.08) and indicate potential
underpredictions of $PM_{2.5}$ BCONs, especially in the free troposphere. Finally, there are
uncertainties associated with MODIS retrievals. Remer et al. (2005) found that the uncertainty of
level 3 MODIS monthly AODs can be up to ±0.05±0.15AOD over the land due to clouds and
surface reflectance. More AOD data from other satellites or AERONET might be considered in
the future work to provide more robust ensemble type of evaluation for AOD.

Figures 3 and 4 compare the 5-year average spatial distribution of major cloud and cloud

radiative variables for the satellite retrievals and two-way WRF-CMAQ simulations, and Table 1
summarizes the domain-average model performance statistics. As shown in Figure 3, WRF-
CMAQ tends to largely underpredict CDNC, COT, and CWP over the whole domain with the
domain-average NMBs of -82.1%, -80.1%, and -51.2%, respectively. Despite the large
underprediction of those cloud variables, the spatial correlations are generally predicted well,
especially for COT and CWP with Rs of 0.84 and 0.79, respectively. Compared to the other





cloud variables, CF is much better predicted with an NMB of -12.2% and an R of 0.92, which is
consistent with the performance reported in Yu et al. (2014). The model can reproduce the high
CFs over northern and northeastern part of domain as well as over oceans while capturing the
low CFs over the mountainous and plateau regions in the U.S. and Mexico. In addition to the
underprediction of PM$_{2.5}$ (thus underestimating CCN), the large underpredictions of cloud
variables (especially CDNC and COT) can be attributed to uncertainties in aerosol microphysics
schemes (Yahya et al., 2016) as well as missing aerosol indirect effects on subgrid convective
clouds (Yu et al., 2014). Gantt et al. (2014) and Zhang et al. (2015b) also showed the aerosol
activation scheme (i.e., Abdul-Razzak and Ghan, 2000) used in the current version of WRF-
CMAQ may have underestimated CDNC and thus CWP and COT due to some missing processes
such as insoluble aerosol adsorption and giant cloud condensation nuclei. Overall, the relatively
poor model performance for cloud variables reflects current limitations in representing aerosol
indirect effects and aerosol-cloud interactions in state-of-science online coupled models. Further
model improvements that incorporate new knowledge from emerging studies should be
conducted in the future.

As shown in Figure 4, WRF-CMAQ predictions of SWCF and LWCF agree well with the

satellite-based values. The model partially captures the elevated SWCF and LWCF over the
Atlantic Ocean, Pacific Northwest, and widespread areas over the eastern U.S. The domain-
average NMBs are -26.0% for SWCF and -22.2% for LWCF, respectively. As discussed earlier,
the underpredictions of SWCF may partially contribute the overprediction of SWDOWN (more
shortwave radiation reaching the ground) and those of LWCF may further lead to the
overpredictions in OLR (more longwave radiation emitted into the space). The performance of
SWCF and LWCF is consistent with the 12-km simulation reported in Yu et al. (2014) and even





slightly better in terms of NMBs, which might be associated with the long-term vs. short-term
simulations. It is also worth noting that SWCF (LWCF) is calculated as the difference between
the clear-sky and all-sky shortwave (longwave) radiation at the top of atmosphere, and so
performance for SWCF and LWCF depends on performance for both radiation and cloud
properties. The generally better performance in terms of model bias for SWCF and LWCF
compared to the cloud variables seems to be driven by the relatively good performance of
shortwave/longwave radiation in the model.
**3.2 Chemical evaluation**
**3.2.1 $O_3$**

Figure 5a shows the spatial distribution of simulated average daily maximum 8-h $O_3$ from

two-way WRF-CMAQ overlaid with observations from both the AIRS-AQS and CASTNET
networks. WRF-CMAQ shows good performance by capturing the spatial distribution of max 8-
h $O_3$ over widespread areas of the domain. The model tends to overpredict $O_3$ along coastlines in
the southeastern U.S., Gulf of Mexico, and Pacific coast, which can be attributed to a poor
representation of coastal boundary layers (Yu et al., 2007), the warm T2 biases as shown in
Figure 1, and lack of $O_3$ sink via halogen chemistry (Sarwar et al., 2015) and deposition to water
(Gantt et al., 2017). The simulation also underpredicts $O_3$ in widespread areas in the Midwest,
eastern, and mountainous regions of the U.S., which is consistent with the results of 36-km
simulations from Wang and Zhang (2012) that used an earlier version of CMAQ v4.6 with the
same CB05 gas-phase mechanism. In addition to cold T2 biases over those areas (Figure 1), the
underpredictions are also believed to be associated with inaccurate representations of precursor
emissions and elevated/complex terrain due to the coarse grid spacing of 36-km over those





regions. Wang and Zhang (2012) found that their 12-km simulation showed improved
performance over similar regions.

Figure 5c shows the monthly variation of domain-average 5-year average $O_3$ mixing

ratios between observations from AIRS-AQS and simulations from two-way WRF-CMAQ, and
Figure 5d shows the diurnal variation of domain-average 5-year average hourly $O_3$ mixing ratios
between observations from CASTNET and simulations from two-way WRF-CMAQ for
representative winter (DJF and blue color) and summer (JJA and red color) seasons. As shown in
Figure 5c, the $O_3$ mixing ratios are overpredicted throughout the year, which is consistent with
overprediction of T2 (figure not shown). The largest overprediction occurs in the relatively cold
months such as September to December. It is interesting that the observations show the largest
monthly $O_3$ mixing ratios in spring and early summer while the simulation shows the peak
during the summer. The difference in timing of peak $O_3$ between observations and simulations
during the year might be associated with uncertainties in the BCONs of $O_3$ that reflect impacts of
the long-range transport and associated stratosphere-troposphere exchange of $O_3$. As shown in
Figure 5d, WRF-CMAQ tends to overpredict $O_3$ during most hours (i.e., 2:00-18:00) in summer
and throughout the whole day in winter partially due to the overprediction of T2, especially in
winter (figure not shown). The diurnal pattern of $O_3$ is captured much better during summer with
much less prediction bias, especially during the nighttime, indicating that the model does a better
job in predicting the evolution of nocturnal boundary layer and atmospheric chemistry in the
warm season than the cold season. The overall overpredictions in this work are also consistent
with previous studies (Eder and Yu, 2006; Appel et al., 2007; Wang et al., 2012), although our
results show much better nighttime performance owing to the application of the ACM2 scheme
that treats both local and non-local closure (Pleim, 2007). As also shown in Table 2, the domain-


average NMBs and NMEs for max 8-h $O_3$ are 12.6% and 13.1% against AIRS-AQS and 1.5%
and 8.4% against CASTNET, respectively. The statistics are also consistent with previous
studies using the CMAQ model (Zhang et al., 2009a; Appel et al., 2013, 2017; Penrod et al.,
2014) and can be considered as good performance according to the criteria suggested by Zhang
et al. (2013) and Emery et al. (2017).
**3.2.2 Aerosols**

Figure 6a shows the spatial distribution of simulated 5-year average $PM_{2.5}$ from two-way

WRF-CMAQ overlaid with observations from both the CSN and IMPROVE networks, and
Figure S1 shows the spatial distribution of the major $PM_{2.5}$ constituents overlaid with
observations from the CSN and IMPROVE network and $PM_{10}$ overlaid with observations from
the AQS network. As shown, WRF-CMAQ performs well for $PM_{2.5}$ over widespread areas of the
Midwest and northeastern U.S., while $PM_{2.5}$ is underpredicted over the southeastern and western
U.S. The model also misses some hot spots of observed concentrations in the western U.S.,
which are mainly caused by TC underpredictions (Figure S1) that are likely linked to poorly
allocated and underestimated wildfire emissions in the NEI (Wiedinmyer et al., 2006; Roy et al.,
2007; Kelly et al., 2019). The relatively large underpredictions over the eastern U.S. are mainly
caused by the combined effects from $SO_4^{2-}$, $NH_4^+$, and TC. As shown in Figure S1, WRF-CMAQ
largely underpredicts $SO_4^{2-}$ in the Midwest and southeastern U.S. mainly due to the
underprediction of oxidants such as $O_3$ (see Figure 5a) (which leads to less production from the
gaseous oxidation), overprediction of precipitation (see Figure 1d) (which leads to more wet
deposition and removal), and large underprediction of cloud fields (see Figure 3) (which leads to
less aqueous phase formation), over the same area. On the other hand, $NH_4^+$ and $NO_3^-$ are either
underpredicted or overpredicted, respectively, over the similar areas mainly due to



underprediction of $SO_4^{2-}$. According to the aerosol thermodynamics, when $SO_4^{2-}$ is
underpredicted, $NH_4^+$ tends to be underpredicted due to its major role as cation. More gaseous
$NH_3$ will be available to neutralize $NO_3^-$, thus leading to overprediction of $NO_3^-$ especially over
the sulfate poor regions (West et al., 1999). Other potential reasons include the inaccurate
assumptions in the thermodynamic module (for example, the internally mixed aerosol state and
equilibrium assumption may not be representative over some regions and different time periods,
S. Yu et al., 2006), uncertainties in emissions of key species such as $NH_3$ and non-volatile
cations that affect particle acidity (Mebust et al., 2003; Wang and Zhang, 2014; Vasilakos et al.,
2018; Pye et al., 2020), and measurement errors especially for $NO_3^-$ and $NH_4^+$ (X.-Y. Yu et al.,
2006; Karydis et al., 2007; Wang and Zhang, 2012). TC underpredictions over most sites of the
domain can be attributed to the underprediction of emissions (e.g., wildfire and primary OC) and
underestimation of secondary organic aerosol (SOA) formation (Appel et al., 2017; Pye et al.,
2017) since EC (a chemically inert species) is overpredicted, which suggest that atmospheric
mixing did not drive the TC underpredictions. Figures 6e-6h show the scatter plots of major
$PM_{2.5}$ components such as $SO_4^{2-}$, $NH_4^+$, and $NO_3^-$, and TC. The WRF-CMAQ predicts $PM_{2.5}$
constituents well with the majority of data within the 1:2 ratio lines. Systematic underpredictions
of $SO_4^{2-}$ and $NH_4^+$ and overpredictions of $NO_3^-$ are shown, which are consistent with their spatial
distributions. Relatively large under- and overpredictions of TC compensate each other and lead
to relatively low overall model biases. As also shown in Figure S1, the model fails to reproduce
high concentrations of $PM_{10}$ (those > 20 $\mu g\ m^{-3}$) over widespread areas of the domain, especially
over dust source areas in CA, AZ, and NM. Hong et al. (2017) found the similar large
underprediction of dust using CMAQ v5.0.2 over China and attributed it to a too-high threshold
for friction velocity in the current dust module (Dong et al., 2016). Sea-salt also seems to be





underpredicted by WRF-CMAQ, although sea-salt predictions are better than dust as shown
along the coastlines.
Figures 6c and 6d show the monthly variation of 5-year average $PM_{2.5}$ between
observations from CSN and IMPROVE, respectively, and simulations from two-way WRF-
CMAQ. Both observations and WRF-CMAQ show higher monthly $PM_{2.5}$ concentrations at CSN
sites than IMPROVE sites throughout the year because most CSN sites are in more polluted
urban areas while IMPROVE sites are in rural areas and national parks. The model tends to
underpredict $PM_{2.5}$ over both CSN and IMPROVE sites in the warm months (i.e., April to
September) mainly due to the underpredictions of $SO_4^{2-}$ and OC while it overpredicts $PM_{2.5}$ in
cold months mainly due to $NO_3^-$. The model also captures the seasonality of $PM_{2.5}$ better over
CSN sites than IMPROVE sites, especially in the summer months. The large underpredictions
over IMPROVE sites during summer months are likely due to the underestimation of precursor
emissions (such as wildfire emissions).
There are no universally accepted performance criteria for aerosols. As recommended by
some previous studies (Zhang et al., 2006; Wang and Zhang, 2012), generally ±15% and ±30%
for model biases and 30% and 50% for model errors can be considered as good and acceptable
performance. As shown in Table 2, WRF-CMAQ in this work demonstrates an overall good or
acceptable performance in predicting aerosols in terms of statistics especially for $PM_{2.5}$, $NO_3^-$,
$NH_4^+$, and TC. It shows the domain-average NMBs of -7.0% and -13.7% for $PM_{2.5}$ against CSN
and IMPROVE, respectively; NMBs of -26.7% and -27.2% for $SO_4^{2-}$ against CSN and
IMPROVE, respectively; NMBs of 16.6% and 14.6% for $NO_3^-$ against CSN and IMPROVE,
respectively; an NMB of -14.3% for $NH_4^+$ against CSN; NMBs of 20.6% and 29.4% for EC
against CSN and IMPROVE, respectively; an NMB of -28.9% for OC against IMPROVE; and





NMBs of -9.4% and -9.2% for TC against CSN and IMPROVE, respectively. The relatively
large underpredictions of $PM_{10}$, i.e., an NMB of -45.9% against AQS, indicate further
improvements of dust emissions are warranted. Overall, the aerosol performance is also
comparable or better than previous CMAQ or WRF-CMAQ applications (Wang and Zhang,
2012; Penrod et al., 2014; Yu et al., 2014). For example, Penrod et al. (2014) showed 5-year
(2001-2005) summer mean NMBs of -19.1% to -17.6% for $PM_{2.5}$ against CSN and IMPROVE
data over the CONUS using the CMAQ v5.0 and Yu et al. (2014) reported the monthly mean
NMBs of -6.2% and -16.8% for $PM_{2.5}$ against CSN and IMPROVE over the eastern U.S. using
the same version of WRF-CMAQ as that used in this study.
**3.2.3 Column abundance**
Figure 7 shows the spatial distribution of 5-year average column abundances between
various satellite products and two-way WRF-CMAQ for column CO, TOR, column $NO_2$, and
column HCHO, and Table 2 summarizes the statistics. As shown, WRF-CMAQ can reproduce
the spatial distribution of the column abundances of gases quite well with Rs ranging from 0.83
to 0.91. TOR, column $NO_2$ and column HCHO are also generally well predicted in terms of
magnitude with NMBs of 1.6%, -14.5%, and 18.0%, respectively. Systematic underpredictions
for column CO occur over the whole domain with an NMB of -26.6% for a few reasons. First,
the BCONs of CO may be significantly underestimated from the CESM model. Using
WRF/Chem or its variant, Zhang et al. (2016b, 2019) found that the column CO performance
could be greatly improved by adjusting the BCON using the satellite observation. A similar
approach could be applied in future WRF-CMAQ simulations as well. Second, as pointed by
Heald et al. (2003), the regional emissions, especially biomass burning, could be a significant
source for elevated CO concentrations and thus underestimation of these emissions could





contribute to the CO underprediction. A more robust set of fire emissions from FINN generated
by NCAR based on satellite retrievals has been applied to the similar time period recently but
using the WRF-Chem model (Zhang and Wang, 2019) and were found to improve the CO
performance. Last, Emmons et al. (2009) showed positive biases (i.e., 19%) of MOPITT
retrievals over the land when compared to in-situ measurements and the biases may have been
increasing over time due to the MOPITT bias drift (e.g., 0.5% $yr^{-1}$ for version 7 retrieval). The
predicted TOR can capture the observed high values over the eastern U.S. and oceans and the
low values in elevated terrain; and it shows the best performance among all gas species. Both
satellite observations and simulations can capture the elevated column $NO_2$ over the industrial
and metropolitan areas in the domain where large nitrogen oxide ($NO_x$) emission sources are
located. The model shows moderate underprediction which can be attributed to both
uncertainties in the emissions and satellite retrievals. For example, the lightning emissions of
$NO_x$ are missing from this study, which have been found by previous studies (Allen et al., 2012)
to contribute up to $2.0 \times 10^{15}$ molecules $cm^{-2}$ over the southern U.S., the Gulf of Mexico, and
northern Atlantic Ocean during certain episodes. Boersma et al. (2004) also found that different
column $NO_2$ retrieval approaches may lead to large errors (> 25%) over polluted areas. Column
HCHO over the CONUS especially the southeastern U.S. is well predicted in terms of magnitude
and spatial distribution and correlates well with the biogenic emission source regions. The
underprediction of column HCHO may thus indicate potential underestimation of biogenic
emissions from the BEIS. Other reasons including potential low yield of HCHO from isoprene
and terpene in the CB05 mechanism and uncertainties in satellite retrievals (Stavrakou et al.,
2009; Lorente et al., 2017)
**3.2.4 Simulated $O_3$ and $PM_{2.5}$ exceedances of NAAQS levels**





National Ambient Air Quality Standards (NAAQS) are set for criteria pollutants,
including $O_3$ and $PM_{2.5}$, to provide protection against adverse health and welfare effects
(www.epa.gov/criteria-air-pollutants/naaqs-table). In this section, the average number of days
per year where the 24-hr $PM_{2.5}$ NAAQS level (35 µg m$^{-3}$) and the max 8-h $O_3$ NAAQS level (70
ppb) are exceeded from the WRF-CMAQ predictions is compared with the number of
exceedances in the monitoring data (i.e., $O_3$ from AQS and CASTNET and $PM_{2.5}$ from
IMPROVE and CSN). This comparison is intended to better characterize the ability of the model
to simulate the high-concentration days that could be especially relevant in regulatory
assessments. In Figure 8, the five-year average of the annual number of exceedance days is
shown for WRF-CMAQ and the monitoring data at monitor locations. The sizes of circles and
shades of color represent the magnitude of exceedances (i.e., larger circles and darker shades
indicate a greater number of exceedance days).  As shown, the observations indicate a large
number of annual exceedance days for max 8-h $O_3$ over major cities, especially in CA, TX, the
Midwest, and northeastern U.S. The spatial distribution of the observed number of exceedance
days from the AQS and CASTNET networks aligns well with the nonattainment map reported by
the Green Book of U.S. EPA (https://www.epa.gov/green-book). The WRF-CMAQ model also
generally captures the distribution of the number of exceedance days very well, especially in CA.
The domain-average values of NMB, NME, and R are -3.4%, 14.0%, and 0.98, respectively, also
indicating a good performance. For $PM_{2.5}$, the largest number of exceedance days based on the
IMPROVE and CSN observations mainly occurs in the northwestern U.S., Midwest, and major
cities in the northeastern U.S. The number of exceedance days is generally much lower for $PM_{2.5}$
than $O_3$. The spatial distribution of the number of exceedance days for observed $PM_{2.5}$ aligns
well with nonattainment areas reported by the Green Book from U.S. EPA in CA. However, the





number of simulated PM$_{2.5}$ exceedance days underpredicts the observation-based values in the
western U.S. mainly due to large underpredictions of PM$_{2.5}$ concentrations in the same areas as
shown in Figure 6a.  The simulation better predicts the distribution of the number of exceedance
days in the eastern U.S. where terrain is relatively flat and wildfire less prevalent.  The domain-
average values of NMB, NME, and R are -29.0%, 80.8%, and 0.21, respectively.
**4. Impacts of chemistry-meteorology feedbacks**

In this section, the impacts of chemistry-meteorology feedbacks including aerosol direct

and indirect effects on regional meteorology and air quality over the U.S. are further examined
by comparing results from two-way WRF-CMAQ and offline coupled WRF and CMAQ. Model
performance from the two sets of simulations is first compared to demonstrate the potential
performance improvements of the two-way model, and the impacts on regional meteorology and
air quality are further investigated via the spatial difference plots for selected variables and
species.
**4.1 Meteorology**

Figures 1 and 4 compare observations and simulations from the two-way WRF-CMAQ

and WRF-only models for precipitation and SWCF/LWCF, respectively. Table 1 also
summarizes the model performance statistics for all major meteorological variables for the two
simulations. The statistics of some cloud variables from the WRF-only simulation are not
available due to missing model outputs.  Overall, good performance is evident for both
simulations for surface meteorological variables with slightly better performance (except for
RH2) for the two-way WRF-CMAQ simulation than the WRF-only simulation. The MBs for the
two-way WRF-CMAQ vs. WRF-only simulation are 0.1°C vs 0.2 °C for T2, 2.2% vs 1.8% for





RH2, 0.44 m s$^{-1}$ vs 0.46 m s$^{-1}$ for WS10, 32.8 degree vs 33.4 degree for WD10, and 0.14-0.71
mm day$^{-1}$ vs 0.2-0.8 mm day$^{-1}$ for precipitation. The spatial distributions for SWCF and LWCF
are slightly better captured especially over the Midwest, Atlantic Ocean, and Pacific Northwest
regions. Compared to WRF-only, two-way WRF-CMAQ shows noticeably better performance in
terms of both MB and RMSE for radiation and cloud forcing, with MBs of 37.0 vs. 24.2 W m$^{-2}$
for SWDOWN, 28.5 vs 17.6 W m$^{-2}$ for GSW, -10.6 vs. -6.1 W m$^{-2}$ for GLW, 2.8 vs. 2.0 W m$^{-2}$
for OLR, -17.6 vs. -10.7 W m$^{-2}$ for SWCF, and -5.9 vs. -5.3 W m$^{-2}$ for LWCF. These results are
consistent with those reported by Yahay et al. (2015a,b) that showed similar improvements in
meteorological and radiative variables when comparing predictions from WRF-Chem with those
from WRF only.  Since identical inputs and physics options are used in both simulations, the
differences in performance for meteorological variables is due to the consideration of feedback
processes among chemistry, aerosol, cloud, and radiation in the two-way coupled WRF-CMAQ
simulation.

Figure 9 shows the difference plots of selected major meteorological variables including

SWDOWN, T2, RH2, WS10, PBL height, and precipitation between two-way WRF-CMAQ and
WRF-only. As shown, the incoming shortwave radiation is reduced by up to 24.8 W m$^{-2}$ (13.6%)
with a domain-average of 13.0 W m$^{-2}$ (6%) due to the combined aerosol direct and indirect
radiative effects over the domain. The reduction is predominant over the eastern U.S. where both
aerosol loading and cloud cover are high and over the oceans where cloud cover is high. The
magnitude of shortwave radiation reduction in this work is consistent with other studies. For
example, Wang et al. (2015) found that the combined aerosol direct and indirect effects using the
WRF/Chem model, which includes the sub-scale cloud forcing not treated in the current WRF-
CMAQ model, may decrease the incoming shortwave radiation by 16.0 W m$^{-2}$ in the summer





over the U.S. Hogrefe et al. (2015) reported the reduction of shortwave radiation may reach up to
20 W m$^{-2}$ over the eastern U.S. by only considering the aerosol direct effect using an older
version of WRF-CMAQ v5.0.1. Xing et al. (2015b) showed that the aerosol direct forcing may
cause the surface shortwave radiation to decrease by up to 10 W m$^{-2}$ over the eastern U.S. over a
decadal time period using WRF-CMAQ v5.0. The reduction of shortwave radiation further
reduces the surface temperature by up to 0.25 °C over the eastern U.S., which is much larger
than the reduction of 0.1 °C reported by Hogrefe et al. (2015), mainly due to the inclusion of
aerosol indirect effects. However there are smaller reductions of T2 over the Pacific Ocean and
even increases (by up to 0.1 °C) over large areas of Atlantic Ocean and Gulf of Mexico where
much larger reductions of shortwave radiation occur. As pointed by Wang et al. (2015), due to
the much larger heat capacity of ocean, the response of sea surface temperature is less sensitive
to the change of shortwave radiation for ocean compared to the land. The large increase of
incoming longwave radiation and latent heat (figures not shown) caused by the aerosol indirect
effects and other complex feedback processes over the ocean compensates for the reduction of
shortwave radiation, especially over the Atlantic Ocean and Gulf of Mexico, and thus leads to
less reduction or even increases of T2. RH2 is found to mostly increase by 3.4% over the land
caused by the decrease of temperature while decrease by 2.6% over the ocean caused by either
the increase of temperature or large decrease of water vapor. Over the land, the decreases in solar
radiation and T2 along with the latent heat (figure not shown) lead to a more stable PBL and thus
suppress the wind (by reducing the wind speed as shown). Over the ocean, the changes lead to a
more unstable PBL and thus enhance the wind over the ocean. The wind speed and PBL height
are reduced by up to 0.05 m s$^{-1}$ and 25 m, respectively, over the U.S. The aerosol feedbacks on
precipitation are also mixed with relatively large decreases by up to 0.4 mm day$^{-1}$ over the U.S.



and increases by up to 0.4 mm day$^{-1}$ over oceans. The suppression of precipitation over the land
is mainly due to the formation of more small sized CCNs caused by aerosol indirect effects and
align well with areas with high aerosol loadings while the enhancement of precipitation,
especially along coastlines and over oceans, might be associated with the larger CCN formation
via more activated sea-salt particles as indicated by Zhang et al. (2010) and Wang et al. (2015).
**4.2 Air Quality**
Figures 5 and 6 compare observations and simulations from two-way WRF-CMAQ and
offline CMAQ for $O_3$, $PM_{2.5}$, and $PM_{2.5}$ constituents. Table 2 summarizes the statistics for all
major chemical variables for the two simulations. As shown in Figure 5, two-way WRF-CMAQ
shows better performance for both the monthly variation of $O_3$ (throughout the whole year) over
AQS sites and the diurnal pattern of $O_3$ (especially during winter) over CASTNET sites due to
better performance of T2 and radiation compared to offline WRF and CMAQ. As shown in
Figure 6, two-way WRF-CMAQ shows similar spatial distribution of $PM_{2.5}$ and better
performance for $PM_{2.5}$ for most of months over CSN sites and for cold seasons across
IMPROVE sites compared to offline CMAQ. It also shows systematically better performance for
$SO_4^{2-}$, $NO_3^{-}$, $NH_4^{+}$, and TC with more data within 1:2 and closer to 1:1 ratio lines of scatter plots.
Overall, as shown in Table 2, both simulations show generally good performance for all major
chemical species except for $PM_{10}$. For example., the domain-average NMBs are 12.6% (AQS)
and 1.5% (CASTNET) vs. 17.7% (AQS) and 7.7% (CASTNET) for $O_3$ and -7.0% (CSN) and -
13.7% (IMPROVE) vs. -3.4% (CSN) and -5.7% (IMPROVE) for $PM_{2.5}$ for two-way WRF-
CMAQ and offline-coupled CMAQ, respectively. The two-way WRF-CMAQ shows better
domain-wide statistics in terms of both correlation and biases for many variables including $O_3$,
$SO_4^{2-}$, $NO_3^{-}$, $NH_4^{+}$, and EC as well as TOR and column $NO_2$, apparently due to the treatment of





chemistry-meteorology feedbacks. Offline CMAQ performs better for total $PM_{2.5}$ especially in
the western U.S. due to higher dust emissions from higher wind speed and higher SOA due to
stronger radiation and higher temperature. However more robust comparisons are needed in the
future with improved dust emissions and the use of FINN wildfire emissions.

Figure 10 shows the difference plots of selected chemical variables including CO, $O_3$,

$NO_x$, volatile organic compounds (VOCs), $SO_4^{2-}$, SOA, $PM_{2.5}$, and $PM_{10}$ between two-way
WRF-CMAQ and offline-coupled CMAQ. As shown, the CO mixing ratios decrease by up to
79.2 ppb (27.8%) especially over the western U.S. with a domain-average reduction of 3.0 ppb
(3.1%) due to reduced formation of CO from the oxidation of VOCs caused by reduced solar
radiation as indicated by Zhang et al. (2017). Such reductions seem to dominate over the
increases caused by reduced PBL height, especially in the western U.S. where PBL height
reductions are minimum. The $O_3$ mixing ratios decrease by up to 5.2 ppb (16.2%) with domain-
average of 1.7 ppb (4.2%) mainly due to the reduced solar radiation and T2. The change of $O_3$ is
consistent with other studies such as Makar et al. (2015) and Wang et al. (2015) that also
reported lower $O_3$ mixing ratios caused by aerosol direct and indirect effects. On the other hand,
both $NO_x$ and VOC mixing ratios increase over the eastern U.S. while they decrease over the
western U.S. The increase should be caused by the combination of the large reduction of PBL
mixing and reduced solar radiation which reduces $NO_2$ photolysis and VOC oxidation to SOA.
For aerosol species, $SO_4^{2-}$ concentrations increase by up to 0.38 μg m$^{-3}$ (26.6%) especially over
the eastern U.S.  In fact, the reduction of $O_3$ mixing ratios due to aerosol effects is expected to
reduce $SO_4^{2-}$ production via the gas-phase oxidation pathway due to the influence of $O_3$ on OH,
but increase $SO_4^{2-}$ production via the aqueous-phase chemistry pathway due to more clouds in
the two-way WRF-CMAQ simulation. Thus, the net increase of $SO_4^{2-}$ is more dominate by the





aqueous-phase chemistry instead of the gas-phase oxidation. This net increase of $SO_4^{2-}$, in turn,
leads to an increase of $NH_4^+$ and decrease of $NO_3^-$ (figures not shown) through aerosol
thermodynamic equilibrium. SOA concentrations decrease by up to 0.34 µg m$^{-3}$ (41.6%)
especially over the eastern U.S. due to the large reduction of oxidants. PM$_{2.5}$ concentrations also
decrease by up to 5.2 µg m$^{-3}$ (49.1%) with a domain-average of 0.34 µg m$^{-3}$ (8.6%), and PM$_{10}$
concentrations decrease by up to 19.3 µg m$^{-3}$ (64.8%) with a domain-average of 1.1 µg m$^{-3}$
(11.1%). The reductions are more apparent over the western U.S. than the eastern U.S. partially
due to the compensation of the increase of $SO_4^{2-}$ and $NH_4^+$ and decrease of other secondary
aerosols over the eastern U.S., as well as the relatively large reduction of dust concentrations
over the western U.S. caused by reduced wind speed.
**5. Summary and conclusion**
In this study, two sets of long-term simulations for 2008-2012 using the two-way coupled
WRF-CMAQ and offline coupled WRF and CMAQ, respectively, are conducted, evaluated, and
compared to investigate the performance improvements due to chemistry-meteorology feedbacks
and impacts of those feedbacks on the reginal air quality in the U.S. First, the two-way coupled
WRF-CMAQ simulation with both aerosol direct and indirect radiative forcing is
comprehensively evaluated. The results show that WRF-CMAQ performs well for major surface
meteorological variables such as temperature at 2 m, relative humidity at 2 m, wind speed at 10
m, and precipitation with domain-average MBs of 0.1 °C, 2.2 %, 0.44 m s$^{-1}$, and 0.14-0.28 mm
day$^{-1}$ (except for 0.71 mm day$^{-1}$ against NCDC), respectively. The overall small warm bias
compared to other studies is most likely associated with the soil moisture nudging technique used
in the PX land surface scheme. The relatively large positive biases for precipitation are found to
be more apparent when observed precipitation is low (dominated more by the non-convective





precipitation) and are thus believed to be more associated with uncertainties in the Morrison
microphysics scheme. The long-term simulation also shows generally good performance for
major radiation and cloud radiative variables. Relatively large model biases still exist for cloud
variables such as CDNC, COT, and CWP, indicating that the processes associated with aerosol
indirect effects are still not well understood and an accurate simulation of those effects is still
challenging using state-of-the-science models.
Two-way WRF-CMAQ also shows generally good or acceptable performance for max 8-
h $O_3$, $PM_{2.5}$ and $PM_{2.5}$ constituents, with NMBs generally within ±15% for $O_3$ and ±30% for
$PM_{2.5}$ species. For example, the domain-average NMBs are 12.6 % and 1.5 % for max 8-h $O_3$
against AQS and CASTNET and -7.0 % and -13.7 % for $PM_{2.5}$ against CSN and IMPROVE,
respectively. $O_3$ mixing ratios are overpredicted for most months, especially in the winter, in part
due to the larger overprediction of T2 during the cold season. The overall model biases are small
for $PM_{2.5}$ due to the compensation of relatively large underpredictions of $SO_4^{2-}$ and OC,
especially in the warm season, and overprediction of $NO_3^-$ in the cold season. In addition to
biases inherited from the meteorology, the model performance for chemistry also suffers from
uncertainties associated with emissions, the use of a coarse spatial resolution, and representation
of aerosol formation pathways in the model. For example, the relatively large biases for EC
might be associated with poorly allocated anthropogenic/wildfire emissions and those for OC
might be due to underestimation of SOA formation in version 5.0.2 of CMAQ. WRF-CMAQ
also predicts the column abundances of chemical species well and the relatively large model
biases for CO are found to be associated with an underestimation of BCONs. The model better
reproduces the observed number of exceedance days for $O_3$ than $PM_{2.5}$ mainly due to better
performance for $O_3$ than $PM_{2.5}$ concentrations.



The performance comparison between two-way WRF-CMAQ and WRF-only simulations
shows that two-way WRF-CMAQ model performs better for major surface meteorological,
radiation, and cloud radiative variables due to the consideration of chemistry-meteorology
feedbacks associated with aerosol direct and indirect forcing. The feedbacks are found to reduce
the 5-year average SWDOWN by up to 24.8 W m$^{-2}$, T2 by up to 0.25 °C, PBL height by up to 25
m, wind speed by up to 0.05 m s$^{-1}$, and precipitation by up to 0.4 mm day$^{-1}$ over the CONUS,
which in turn affect the air quality significantly. As a result of feedbacks, two-way WRF-CMAQ
outperforms offline CMAQ for $O_3$, $SO_4^{2-}$, $NO_3^-$, $NH_4^+$, and EC as well as TOR and column $NO_2$
in terms of both spatiotemporal variations and domain-average statistics due to better
meteorology performance for variables such as T2, WS10, radiation, and precipitation. Despite
these improvements, the offline CMAQ performs better for total $PM_{2.5}$ in terms of domain-
average statistics, which could be partially caused by the compensation of larger under- and
over-predictions of $PM_{2.5}$ constituents. More robust comparison for $PM_{2.5}$ should be performed
with improved dust and wildfire emissions in future work. Chemistry-meteorology feedbacks are
found to play important roles in affecting U.S. air quality by reducing domain-wide 5-year
average surface CO by 3.0 ppb (3.1%) and up to 79.2 ppb (27.8%), $O_3$ by 1.7 ppb (4.1%) and up
to 5.2 ppb (16.2%), $PM_{2.5}$ by 0.34 μg m$^{-3}$ (8.6%) and up to 5.2 μg m$^{-3}$ (49.1%), and $PM_{10}$ by 1.1
μg m$^{-3}$ (11.1%) and up to 19.3 μg m$^{-3}$ (64.8%) mainly due to reduction of radiation, temperature,
and wind speed.
In summary, the two-way coupled WRF-CMAQ modeling in this study shows generally
satisfactory and consistent performance for the long-term prediction of regional meteorology and
air quality when compared to other studies in the literature. Possible causes for the
meteorological and chemical biases that were identified through this work can provide valuable



information for future model development to improve the two-way coupled WRF-CMAQ model
and those biases should also be considered when making future climate/air quality projections.
Non-negligible model improvements for many major meteorological and chemical variables
compared to the traditional application of offline coupled WRF and CMAQ suggest the
importance of chemistry-meteorology feedbacks, especially aerosol direct and indirect effects.
The feedbacks should be considered along with other factors in developing future model
applications to inform policy making.
**Code Availability**
The modeling system used in this study is based on the 2-way coupled WRF-CMAQ model
derived from WRF v3.4 and CMAQ v5.0.2. Relevant code for CMAQ v5.0.2, its coupling to
WRF and aerosol direct feedbacks are publicly available from: doi:10.5281/zenodo.1079898.
WRF v3.4 code can be downloaded from
http://www2.mmm.ucar.edu/wrf/users/download/get_source.html. The version of the coupled
WRF-CMAQ model with the additional indirect aerosol forcing approach of Yu et al. (2014) can
be downloaded from the following website: https://person.zju.edu.cn/shaocaiyu#674502.
**Author contribution**
YZ and KW designed the study and all the simulations. SY developed the two-way coupled
WRF-CMAQ code. KW conducted all the simulations and performed the analyses. KW prepared
the manuscript with contributions from all co-authors.
**Competing interests**
The authors declare that they have no conflict of interest.



**Acknowledgements**
This work was developed at North Carolina State University and Northeastern University under
Assistance Agreement No. RD835871 awarded by the U.S. Environmental Protection Agency to
Yale University. The views expressed in this manuscript are those of the authors alone and do
not necessarily reflect the views and policies of the U.S. Environmental Protection Agency. EPA
does not endorse any products or commercial services mentioned in this publication. High
performance computing was support from Yellowstone (ark:/85065/d7wd3xhc) provided by
NCAR's CISL, sponsored by the NSF and the Stampede XSEDE high-performance computing
support under the NSF ACI-1053575. The work of S. Yu is supported by the Department of
Science and Technology of China (No. 2016YFC0202702, 2018YFC0213506 and
2018YFC0213503), National Research Program for Key Issues in Air Pollution Control in China
(No. DQGG0107) and National Natural Science Foundation of China (No. 21577126 and
41561144004). The authors gratefully acknowledge the availability of CERES, GPCP, MODIS,
MOPITT, NCDC, OMI, PRISM, SCHIAMACHY, and TMPA data. The authors thank Dr. Ralf
Bennartz from Vanderbilt University for providing the CDNC data. The authors also would like
to thank Drs. Jerry Herwehe and Shannon Koplitz from the U.S. EPA for their constructive and
very helpful comments.

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





Table 1. The 5-year (2008-2012) average performance statistics for meteorological variables between two-way WRF-CMAQ and WRF-only simulations.

| Variables | Datasets | Mean Obs | Two-way WRF-CMAQ | | | | | WRF-only | | | | |
|---|---|---|---|---|---|---|---|---|---|---|---|---|
| | | | Mean Sim | R | MB | NMB (%) | RMSE | Mean Sim | R | MB | NMB (%) | RMSE |
| T2 (ºC) | | 12.9 | 13.0 | 0.98 | 0.1 | 0.8 | 1.0 | 13.1 | 0.98 | 0.2 | 1.8 | 1.1 |
| RH2 (%) | NCDC | 69.1 | 71.3 | 0.88 | 2.2 | 3.2 | 5.3 | 71.0 | 0.88 | 1.8 | 2.6 | 5.2 |
| WS10 (m s$^{-1}$) | | 3.74 | 4.18 | 0.52 | 0.44 | 11.7 | 1.15 | 4.20 | 0.52 | 0.46 | 12.4 | 1.16 |
| WD10 (deg) | | 154.4 | 187.2 | 0.07 | 32.8 | 21.3 | 47.7 | 187.8 | 0.06 | 33.4 | 21.6 | 48.1 |
| | NCDC | 1.84 | 2.55 | 0.62 | 0.71 | 38.4 | 1.27 | 2.64 | 0.62 | 0.8 | 43.5 | 1.33 |
| | NADP | 2.66 | 2.81 | 0.84 | 0.15 | 5.8 | 0.7 | 2.9 | 0.84 | 0.24 | 9.3 | 0.73 |
| Precipitation (mm day$^{-1}$) | GPCP | 2.15 | 2.43 | 0.79 | 0.28 | 13.0 | 0.9 | 2.45 | 0.80 | 0.30 | 14.1 | 0.9 |
| | PRISM | 2.16 | 2.30 | 0.91 | 0.14 | 6.8 | 0.55 | 2.36 | 0.91 | 0.20 | 9.5 | 0.56 |
| | TMPA | 2.28 | 2.50 | 0.86 | 0.22 | 9.9 | 0.81 | 2.52 | 0.86 | 0.24 | 10.7 | 0.82 |
| SWDOWN (W m$^{-2}$) | | 185.6 | 209.8 | 0.97 | 24.2 | 13.0 | 25.7 | 222.6 | 0.96 | 37.0 | 19.9 | 38.3 |
| GSW (W m$^{-2}$) | | 158.5 | 176.0 | 0.97 | 17.6 | 11.1 | 19.8 | 187.0 | 0.95 | 28.5 | 18.0 | 30.6 |
| GLW (W m$^{-2}$) | CERES | 322.9 | 316.8 | 0.99 | -6.1 | -1.9 | 8.1 | 312.3 | 0.99 | -10.6 | -3.3 | 12.1 |
| OLR (W m$^{-2}$) | | 241.2 | 243.2 | 0.99 | 2.0 | 0.8 | 3.5 | 244.0 | 0.99 | 2.8 | 1.2 | 4.2 |
| SWCF (W m$^{-2}$) | | -41.1 | -30.4 | 0.74 | -10.7 | -26.0 | 13.7 | -23.5 | 0.63 | -17.6 | -42.8 | 20.1 |
| LWCF (W m$^{-2}$) | | 23.7 | 18.4 | 0.73 | -5.3 | -22.2 | 6.5 | 17.8 | 0.74 | -5.9 | -24.9 | 6.9 |
| AOD | | 0.15 | 0.05 | 0.60 | -0.1 | -64.8 | 0.11 | N/A | N/A | N/A | N/A | N/A |
| CF | | 0.57 | 0.50 | 0.92 | -0.07 | -12.2 | 0.09 | N/A | N/A | N/A | N/A | N/A |
| CDNC (cm$^{-3}$) | MODIS | 163.3 | 29.3 | 0.35 | -134.0 | -82.1 | 138.8 | N/A | N/A | N/A | N/A | N/A |
| CWP (g m$^{-2}$) | | 167.4 | 81.6 | 0.79 | -85.8 | -51.2 | 90.4 | N/A | N/A | N/A | N/A | N/A |
| COT | | 15.3 | 3.0 | 0.84 | -12.3 | -80.1 | 12.6 | N/A | N/A | N/A | N/A | N/A |

[*]outputs of AOD, CF, CDNC, CWP, and COT are not available from WRF-only simulations



Table 2. The 5-year (2008-2012) average performance statistics for chemical variables between two-way WRF-CMAQ and offline CMAQ simulations.

| Variables | Datasets | Mean Obs | Two-way WRF-CMAQ | | | | | Offline CMAQ | | | | |
|---|---|---|---|---|---|---|---|---|---|---|---|---|
| | | | Mean Sim | R | MB | NMB (%) | NME (%) | Mean Sim | R | MB | NMB (%) | NME (%) |
| Max 8-hr $O_3$ (ppb) | AQS | 43.5 | 49.0 | 0.66 | 5.5 | 12.6 | 13.1 | 51.2 | 0.66 | 7.7 | 17.7 | 17.9 |
| | CASTNET | 42.2 | 42.8 | 0.65 | 0.6 | 1.5 | 8.4 | 45.1 | 0.65 | 3.0 | 7.0 | 10.5 |
| $PM_{2.5}$ ($\mu g\ m^{-3}$) | CSN | 10.7 | 9.9 | 0.50 | -0.75 | -7.0 | 21.9 | 10.3 | 0.46 | -0.36 | -3.4 | 21.7 |
| | IMPROVE | 4.78 | 4.13 | 0.88 | -0.65 | -13.7 | 26.6 | 4.51 | 0.87 | -0.27 | -5.7 | 23.2 |
| $PM_{10}$ ($\mu g\ m^{-3}$) | AQS | 24.0 | 13.0 | 0.02 | -11.0 | -45.9 | 49.6 | 15.4 | 0.14 | -8.6 | -35.6 | 45.0 |
| $SO_4^{2-}$ ($\mu g\ m^{-3}$) | CSN | 2.32 | 1.70 | 0.88 | -0.62 | -26.7 | 27.1 | 1.57 | 0.89 | -0.75 | -32.3 | 32.3 |
| | IMPROVE | 1.08 | 0.78 | 0.98 | -0.29 | -27.2 | 27.2 | 0.76 | 0.98 | -0.32 | -29.4 | 29.4 |
| $NO_3^-$ ($\mu g\ m^{-3}$) | CSN | 1.29 | 1.51 | 0.85 | 0.22 | 16.6 | 32.8 | 1.73 | 0.85 | 0.43 | 33.5 | 44.9 |
| | IMPROVE | 0.41 | 0.47 | 0.85 | 0.06 | 14.6 | 42.9 | 0.57 | 0.87 | 0.16 | 39.0 | 51.7 |
| $NH_4^+$ ($\mu g\ m^{-3}$) | CSN | 1.03 | 0.88 | 0.86 | -0.15 | -14.3 | 18.6 | 0.87 | 0.85 | -0.16 | -15.7 | 18.7 |
| EC ($\mu g\ m^{-3}$) | CSN | 0.63 | 0.76 | 0.34 | 0.13 | 20.6 | 52.4 | 0.77 | 0.39 | 0.14 | 22.4 | 50.5 |
| | IMPROVE | 0.18 | 0.23 | 0.80 | 0.05 | 29.4 | 50.8 | 0.25 | 0.79 | 0.07 | 37.7 | 55.6 |
| OC ($\mu g\ m^{-3}$) | IMPROVE | 0.97 | 0.69 | 0.59 | -0.28 | -28.9 | 44.8 | 0.74 | 0.58 | -0.23 | -23.8 | 43.4 |
| TC ($\mu g\ m^{-3}$) | CSN | 2.87 | 2.60 | 0.10 | -0.27 | -9.4 | 29.7 | 2.71 | 0.07 | -0.16 | -5.7 | 28.8 |
| | IMPROVE | 0.68 | 0.62 | 0.79 | -0.06 | -9.2 | 37.2 | 0.80 | 0.72 | -0.08 | -9.2 | 39.0 |
| Col. CO ($10^{18}$ mole. $cm^{-3}$) | MOPITT | 1.96 | 1.44 | 0.89 | -0.52 | -26.6 | 26.7 | 1.45 | 0.89 | -0.51 | -26.2 | 26.2 |
| TOR (DU) | OMI | 30.3 | 30.8 | 0.83 | 0.47 | 1.6 | 4.7 | 31.1 | 0.82 | 0.77 | 2.5 | 5.1 |
| Col. $NO_2$ ($10^{15}$ mole. $cm^{-3}$) | SCIAMACHY | 1.27 | 1.09 | 0.91 | -0.18 | -14.5 | 27.1 | 1.08 | 0.91 | -0.19 | -14.9 | 27.3 |
| Col. HCHO ($10^{15}$ mole. $cm^{-3}$) | SCIAMACHY | 5.13 | 4.21 | 0.83 | -0.92 | -18.0 | 20.6 | 4.28 | 0.83 | -0.85 | -16.6 | 19.8 |

Figure 1. Spatial distributions of 5-year average MBs for a) 2-m temperature (T2), b) 2-m
relative humidity (RH2), c) 10-m wind speed (WS10), and d) hourly precipitation from NCDC
for two-way WRF-CMAQ in 2008-2012 and 5-year average of daily precipitation for e) GPCP,
f) PRISM, g) two-way WRF-CMAQ, and h) WRF-only.



Figure 2. Spatial distribution of 5-year average major radiation variables (from top to bottom: SWDOWN, GSW, GLW, OLR, and AOD) between CERES observations (left panel) vs. two-way WRF-CMAQ (right panel) for 2008-2012.



Figure 3. Spatial distribution of 5-year average major cloud variables (from top to bottom: CDNC, CF, COT, and CWP) between MODIS observations (left panel) vs. two-way WRF-CMAQ (right panel) for 2008-2012.



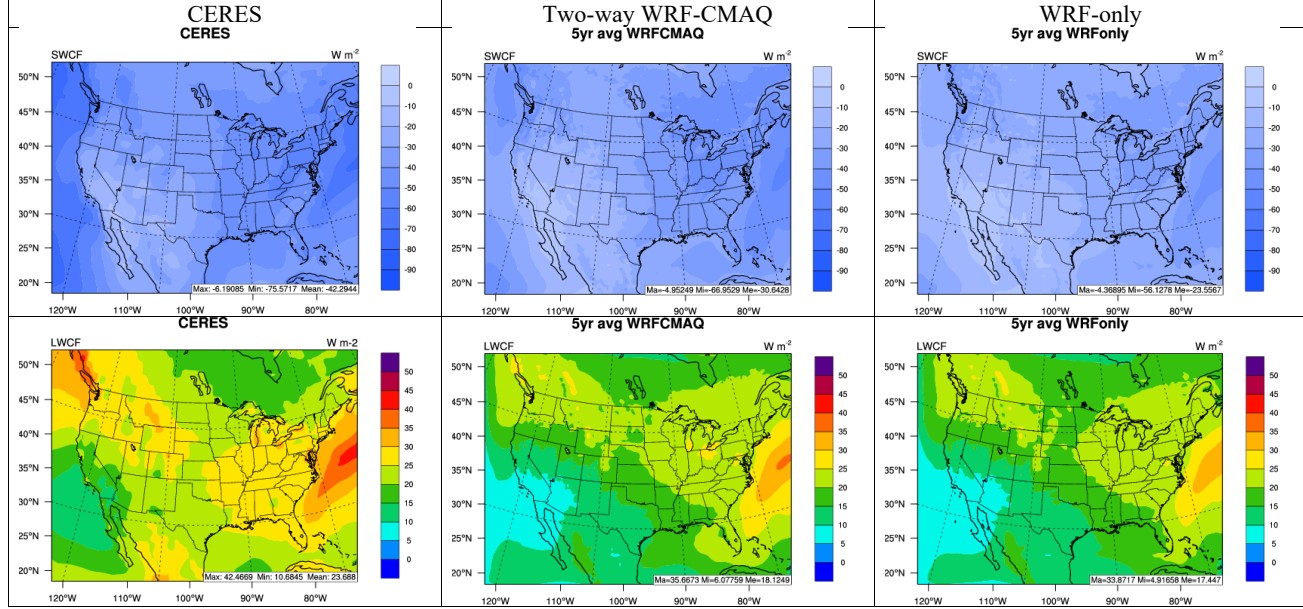

Figure 4. Spatial distribution of 5-year average SWCF (top panel) and LWCF (bottom panel) between SERES observations (left panel) vs. two-way WRF-CMAQ (center panel) and WRF-only (right panel) for 2008-2012.

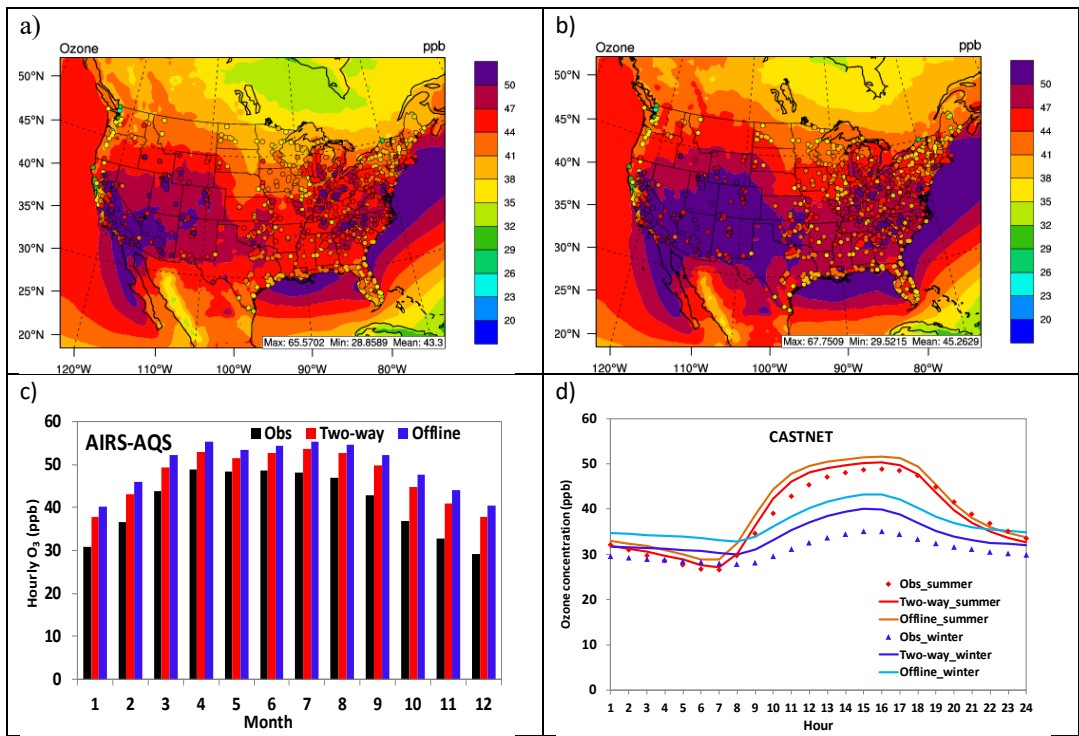

Figure 5. Spatial distributions of 5-year averaged max 8-h $O_3$ overlaid with observations from AIRS-AQS and CASTNET for a) two-way WRF-CMAQ and b) offline CMAQ; c) bar chart for 5-year average monthly $O_3$ between observations (black bar), two-way WRF-CMAQ (red bar), and offline CMAQ (blue bar); and d) diurnal plots of observed (dots) vs. simulated (lines) hourly $O_3$ concentrations against CASTNET for winter (cold colors) and summer (warm colors) in 2008-2012.





Figure 6. Spatial distributions of 5-year averaged daily PM2.5 overlaid with observations from CSN and IMPROVE for a) two-way WRF-CMAQ and b) offline CMAQ; bar charts for 5-year average monthly PM2.5 between observations (black bar), two-way WRF-CMAQ (red bar), and offline CMAQ (blue bar) over c) CSN and d) IMPROVE; and scatter plots of PM2.5 constituents e) $SO_4^{2-}$, f) $NH_4^+$, g) $NO_3^-$, and h) TC) between observations and simulations of two-way WRF-CMAQ (red color) and offline CMAQ (blue) for 2008-2012.





Figure 7. Spatial distribution of 5-year average column abundances (from top to bottom: column CO, TOR, column NO₂, and column HCHO) between various satellite observations (left panel) vs. two-way WRF-CMAQ (right panel) for 2008-2012.



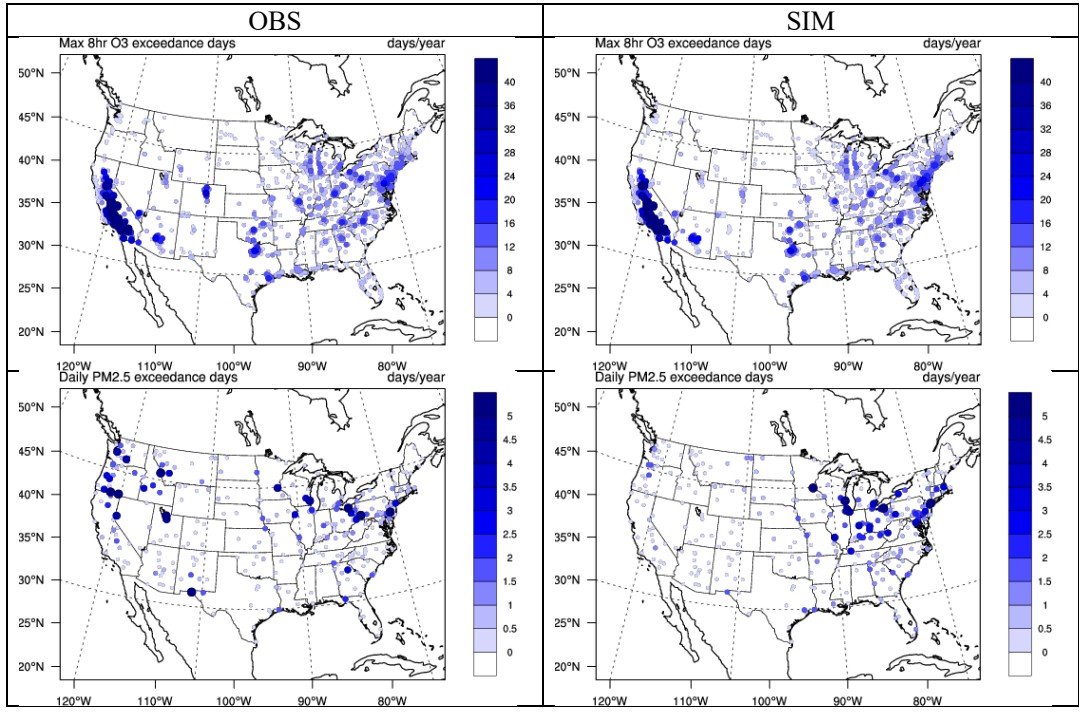

Figure 8. The spatial distribution of 5-year average annual exceedance days of max 8-h $O_3$ and daily $PM_{2.5}$ between observations ($O_3$ over the AIRS-AQS/CASTNET network and $PM_{2.5}$ over the IMPROVE/CSN network) and two-way WRF-CMAQ.

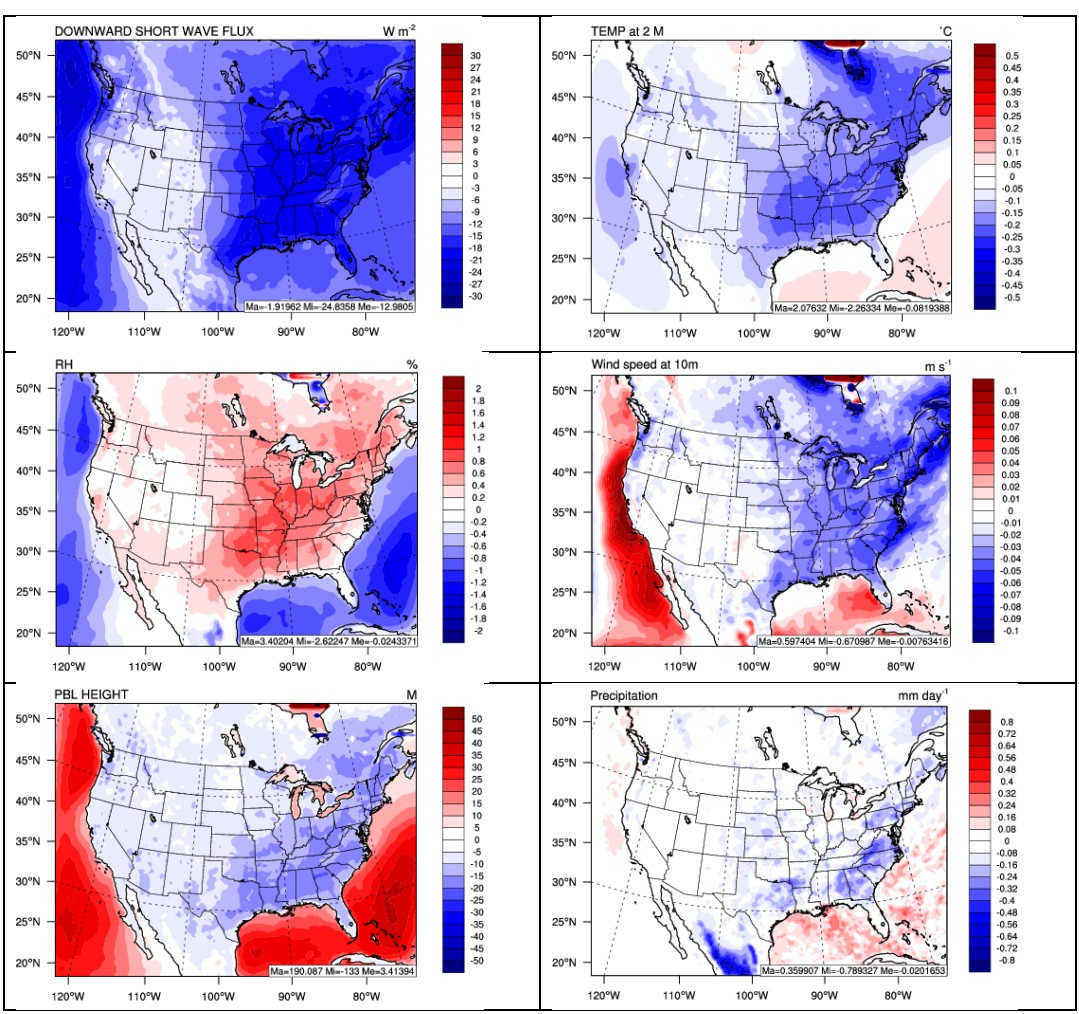

Figure 9. Spatial difference plots (two-way WRF-CMAQ - WRF-only) for major meteorological variables between two-way WRF-CMAQ and WRF-only.





Figure 10. Spatial difference plots (two-way WRF-CMAQ - offline CMAQ) for major chemical species between two-way WRF-CMAQ and offline CMAQ.