# Peer review of "A Comparative Study of Two-way and"

_Geoscientific Model Development, 2020_

## Referee Comment (RC1) · Anonymous Referee #1 · 1 Oct 2020

General Comments:

The manuscript 'A comprehensive study of two-way and offline coupled WRF v3.4 and CMAQ v5.0.2 over the contiguous U.S.: Performance evaluation and impacts of chemistry-meteorology feedbacks on air quality' written by Kai Wang presented the comprehensive comparison of offline (i.e., traditional) CMAQ and two-way coupled CMAQ over the CONUS. To promote our understating on the interaction between meteorology and air quality, the approach of two-way coupled modeling is necessary, and this manuscript can contribute to this purpose. The authors claimed that long-term

simulations on both (two-way and offline) models over the CONUS is important point in this study, because the previous studies have been limited in many aspects (different chemical options, difference meteorological options, or, limited in time to focus on episode analysis). Although I would like to recognize the importance of this study, the evaluation is not well conducted in depth to make the best use of this long-term simulation. Please consider to address the following one major point, and also check the minor comments to improve the manuscript.

Major point:

I would like to disagree the evaluation framework of long-term simulations conducted in this study. The authors stated that "more robust assessments" through five-year simulations; however, the evaluation is only conducted by averaging the five-year dataset. This does not take advantage long-term simulations, and does not provide deep understanding of two-way coupled and offline models comparison. In addition to the averaged field of climatological type data, the comparison should be furthermore focused on trends in five years (if detected from observed facts) or year-to-year variations of both meteorology and air quality. Based on this extended evaluation, it could be finally proved the importance of two-way coupled model. Without such kind of evaluations, this study of long-term simulations will be less important.

Minor points:

1. L26: "modes" is typo of "models"?

2. L178-180 (and abstract): Are this chemical ICON/BCON considered year-to-year variation simulated by CESMv1.2.2/CAM5? Did this model perform well compared to other model(s)? If this model had superiority, please note how this model is important. Without any specific reasons, I feel it is no need to mention this model in the abstract.

3. L183-L185: In my best knowledge, inline dust scheme implemented in CMAQ version 5.0.2 is not the scheme reported by Foroutan et al. (2017) (see, also

https://www.airqualitymodeling.org/index.php/CMAQv5.0.2_Wind_blown_dust_updates).
In addition, this statement contradicts to the discussion in its evaluation (L427-429).
Please address this issue. If the authors implemented the scheme by Foroutan et al.
(2017) in this study, exact explanation is required because this is model development
paper.

4. L212: "PM10" will include PM2.5, hence the expression of "coarse particulate matter" is not appropriate. Or, did the authors calculate PM10-PM2.5 to represent coarse-mode particulate matter?

5. L222 (and related to Section 3.2.3): "paired with the satellite retrievals" means
the deficit grid points in satellite observation are applied for model results? Please
clarify. I guess that some satellite products provide averaging kernel, but how did
the author apply averaging kernel for better comparison between model and satellite
measurements? The detail seems to be dropped here. Please specify.

6. L443-446: The review paper by Emery et al. (2017) (used for ozone evaluation in
this study) also presented the model performance goal/criteria for aerosols. Why these
criteria is not used?

7. Figure 10: For gas species, differences are seen along latitude (approx. each 3-4
deg.) over western U.S.A. and Mexico. What is this difference?

8. Author contribution (L720-723): The contributions of all authors are not explicitly
described here. Is it accepted in this journal style? (see, https://www.geoscientific-
model-development.net/submission.html#manuscriptcomposition)
* * *

---

## Referee Comment (RC2) · Anonymous Referee #2 · 28 Oct 2020

Wang et al. present a study which evaluates the performance of the of Two-way and Offline Coupled WRF v3.4 and CMAQ v5.0.2 over the Contiguous U.S for an extended time period (5 years). Previous works had experimental design deficiencies (e.g., differing physics, chemistry) that his work addresses. The importance of chemical-meteorological feedbacks are increasingly being recognized as essential for the prediction of both weather and atmospheric chemistry, and this work adds well to that body of work. Outside of a major comment w/ regards to the experimental design (cycling between 5-day periods), my main critique of the manuscript is the heavy reliance on

the use of 5-year averages to discuss model performance and comparisons. This is also somewhat related to my major comment about cycling. I think it would benefit the community to examine and discuss seasonal spatial patterns (and thus reasons for model deficiencies), periods of peak aerosol and/or high ozone days (not just number of exceedance, but more details in how the model performs/evolves).

Major comment: Line 185: Are any fields cycled between consecutive 5-day simulations besides chemistry? (e.g., land surface fields?) I think this needs to be discussed in detail how it relates to the experiments. If they are reinitialized every 5 days, should the first day or two be considered in the comparisons? The deviation between the two simulations would likely increase as lead time increases. Here is really comparing 5 years of 5-day forecasts.

Minor comments: Sections 3.1.1 and 3.1.2: It would be much more helpful to at break these comparisons up into summer vs. winter as some biases could be cancelling one another out.

Figure 4: The colors used in the top panel are very hard to distinguish.

3.2.1. Annual average ozone is not really a useful diagnostic, I think showing summer only would be very beneficial.

3.2.2. Again, a seasonal analysis here would be more appropriate (i.e., winter is dominated by NO3, summer with OA (and SO4)).

Figure 5a-b: You could shift the color limits by 20 ppb.

Figure 8. Why not just use more colors instead of the varying dot sizes – hard to distinguish.

Figure 10. Looks to be some weird striping for O3.
* * *

---

## Author Comment (AC1) · 15 Jul 2021

**Responses to Reviewers**

**Anonymous Referee #1**

General Comments:
The manuscript 'A comprehensive study of two-way and offline coupled WRF v3.4
and CMAQ v5.0.2 over the contiguous U.S.: Performance evaluation and impacts of
chemistry-meteorology feedbacks on air quality' written by Kai Wang presented the
comprehensive comparison of offline (i.e., traditional) CMAQ and two-way coupled
CMAQ over the CONUS. To promote our understating on the interaction between meteorology
and air quality, the approach of two-way coupled modeling is necessary, and
this manuscript can contribute to this purpose. The authors claimed that long-term
simulations on both (two-way and offline) models over the CONUS is important point
in this study, because the previous studies have been limited in many aspects (different
chemical options, difference meteorological options, or, limited in time to focus on
episode analysis). Although I would like to recognize the importance of this study, the
evaluation is not well conducted in depth to make the best use of this long-term simulation.
Please consider to address the following one major point, and also check the
minor comments to improve the manuscript.
**Reply: We thank the reviewer's constructive comments and also for recognizing the
importance of this study. We have carefully revised our paper to fully address the reviewer
comments. Please find our point-by-point responses below.**

Major point:
I would like to disagree the evaluation framework of long-term simulations conducted in
this study. The authors stated that "more robust assessments" through five-year simulations;
however, the evaluation is only conducted by averaging the five-year dataset.
This does not take advantage long-term simulations, and does not provide deep understanding
of two-way coupled and offline models comparison. In addition to the averaged
field of climatological type data, the comparison should be furthermore focused
on trends in five years (if detected from observed facts) or year-to-year variations of
both meteorology and air quality. Based on this extended evaluation, it could be finally
proved the importance of two-way coupled model. Without such kind of evaluations,
this study of long-term simulations will be less important.
**Reply: In order to address the reviewer's concern, we have performed the annual trend
analyses and added a new figure (i.e., Figure 3) for major meteorological variables and air
pollutant species. Additional analyses are also added in Sections 3.1.1, 3.2.1, and 3.2.2 in the
revision and attached below (in red color) as well:**

**"Figure 3 shows the bar charts of annual trends for T2, RH2, WS10, and precipitation in
2008-2012. Two-way WRF-CMAQ predicts the annual average T2 very well with MBs <
0.25 °C in all years. The simulation can also capture the increasing trend of T2 from 2008
to 2012 observed by NCDC.  RH2 is consistently overpredicted by the two-way WRF-
CMAQ in all years despite relatively low biases (MBs < 3%). Both observations and
simulations show the lowest RH2 in 2012 and the highest in 2009.  As also shown in Figure
1, the model tends to systematically overpredict both WS10 and precipitation throughout
all years as well. There are no clear trends (i.e., increasing or decreasing) for WS10 and**

precipitation between 2008 to 2012 from either observations or simulations. However two-way WRF-CMAQ is able to capture the lowest wind speed and precipitation both in 2012 and the highest wind speed in 2008 from observations. In general, the model performs very well in reproducing the year-to-year variation for the major meteorological variables between 2008 to 2012."

"Figure 3 also shows the bar charts of annual trends for max 8-h O3 from two-way WRF-CMAQ against AQS and CASTNET observations in 2008-2012. Two-way WRF-CMAQ systematically overpredicts O3 especially against AQS data with MBs typically > 4.0 ppb. The potential reasons for model biases have been discussed earlier in this section. There are no obvious decreasing or increasing trends for max 8-h O3 from AQS or CASTNET observations. However, the model can generally capture the high O3 mixing ratios in 2008 and 2010 and the low O3 mixing rations in 2009 from both AQS and CASTNET. The similar down and up trends between 2008 to 2010 for O3 (i.e., decreasing from 2008 to 2009 and increasing from 2009 to 2010) from AQS observations were also found by Yahya et al. (2016), but not captured by their simulations. Zhang and Wang (2016) was able to reproduce the similar trend over the southeastern U.S. between 2008 to 2010 using their models and attributed the abnormal high 2010 O3 mixing ratios to the extreme dry and warm weather conditions during fall 2010."

"Figure 3 shows the bar charts of annual averaged observations and simulations for PM2.5 over the CSN and IMPROVE sites. Overall, the model performs well for PM2.5 for most of years and better over CSN than IMPROVE sites with general underpredictions in most years. The observations for both CSN and IMPROVE show a general decreasing trend (except for 2010 over CSN) especially over IMPROVE sites. Two-way WRF-CMAQ is able to reproduce the declining trend well particularly over IMPROVE sites and again demonstrate its capability in accurately simulating the year-to-year variations of not only meteorology but air quality."

Overall, our simulations can either capture the decreasing trend for some variables (e.g., T2 and PM$_{2.5}$) or reproduce the year-to-year variation for most of rest variables well, which provide great fidelity in applying this version of two-way coupled WRF-CMAQ model for the future studies.

In addition, we also significantly revised majority of figures and tables (i.e., moving the old Figures 1-7 and Tables 1-2 into supplementary materials and creating the new Figures 1-13 and Tables 1-4) in our revision by adding the seasonal analyses as suggested by the other reviewer which we believe should provide even deeper understanding of the results and further address reviewer's concern here.

Minor points:
1. L26: "modes" is typo of "models"?
Reply: It's indeed modes. Two different coupling modes for the same version of CMAQ model.

2. L178-180 (and abstract): Are this chemical ICON/BCON considered year-to-year

variation simulated by CESMv1.2.2/CAM5? Did this model perform well compared to other model(s)? If this model had superiority, please note how this model is important. Without any specific reasons, I feel it is no need to mention this model in the abstract.

**Reply: Yes. The ICONs/BCONs simulated by CESMv1.2.2/CAM5 are year specific. It's an online coupled global model with many improvements in terms of chemistry and aerosol treatments. The simulations have been comprehensively evaluated against surface, remoting sensing including satellite data, and reanalysis data for major meteorological and chemical variables over Europe, Asia, North America, and the globe. The results show generally satisfactory performance and are also compared with existing global model results such as CESM/CMIP5. More details and model evaluation can be found in He and Zhang (2014) and Glotfelty et al. (2017) as already cited in the original submission.**

**We have added the following statement in the revision (L180-185 in the track-mode version) to cover the above points:**
**"The chemical ICONs/BCONs generated from CESM simulations consider the year-to-year variation. The CESM simulations have been comprehensively evaluated against surface, remoting sensing including satellite data, and reanalysis data for major meteorological and chemical variables over Europe, Asia, North America, and the globe. The results are also compared with other existing global model results and show generally satisfactory/superior performance."**

3. L183-L185: In my best knowledge, inline dust scheme implemented in CMAQ version 5.0.2 is not the scheme reported by Foroutan et al. (2017) (see, also https://www.airqualitymodeling.org/index.php/CMAQv5.0.2_Wind_blown_dust_updates). In addition, this statement contradicts to the discussion in its evaluation (L427-429). Please address this issue. If the authors implemented the scheme by Foroutan et al. (2017) in this study, exact explanation is required because this is model development paper.

**Reply: We thank reviewer for catching this issue. We indeed used the default dust scheme in CMAQv5.0.2, which should be based on Zender et al. (2003) instead of Foroutan et al. (2017). We have fixed this inaccurate citation by updating the reference and now it should be consistent with L427-429 (now L511-513 in track-mode revision).**

4. L212: "PM10" will include PM2.5, hence the expression of "coarse particulate matter" is not appropriate. Or, did the authors calculate PM10-PM2.5 to represent coarsemode particulate matter?

**Reply: The reviewer is right that PM10 evaluated in this work includes PM2.5. So we have rewritten the definition as "particulate matter with diameters of 10 μm or less" at L212 (L222 in track-mode revision). To be consistent, we also redefine PM2.5 as "particulate matter with diameters of 2.5 μm or less" at L208 (L218 in track-mode revision).**

5. L222 (and related to Section 3.2.3): "paired with the satellite retrievals" means the deficit grid points in satellite observation are applied for model results? Please clarify. I guess that some satellite products provide averaging kernel, but how did the author apply averaging kernel for better comparison between model and satellite

measurements? The detail seems to be dropped here. Please specify.

**Reply: Only those grid points with valid satellite observations are considered when paring model results with observations. As noted by the reviewer, averaging kernels (AKs) for some satellite products such as $NO_2$ are only available for level 2 data, however all analyses in this work are based on level 3 data. Also one of previous studies by Schaub et al. (2006) found that both satellite retrievals with or without applying AKs from Global Ozone Monitoring Experiment (GOME), which uses the similar retrieval methods as SCIMACHY used in this study, show generally good agreements with ground-based measured $NO_2$ columns.**

**In short, for the current study, no AKs are applied which may introduce some uncertainties, but won't affect our conclusion. We have acknowledged this issue and also further clarified the data pairing in the revision (L233-236) as below.**

"**Note that only those grid points with valid satellite observations are considered when paring model results with observations and the averaging kernels are not considered when analyzing the column CO and $NO_2$ results, which may introduce some uncertainties**."

6. L443-446: The review paper by Emery et al. (2017) (used for ozone evaluation in this study) also presented the model performance goal/criteria for aerosols. Why these criteria is not used?

**Reply: Thanks for bringing up the criteria proposed in Emery et al. (2017) for aerosols. We have cited Emery et al. (2017) here and removed the sentence "There are no universally accepted performance criteria for aerosols." from the revision. Actually the criteria used in this study are consistent with those recommended in Emery et al. (2017).**

7. Figure 10: For gas species, differences are seen along latitude (approx. each 3-4 deg.) over western U.S.A. and Mexico. What is this difference?

**Reply: It seems to be caused by the WRF-CMAQ interface to deal with feedback interactions among multiple CPUs while conducting parallel computing, which hardly can affect any conclusions.**

8. Author contribution (L720-723): The contributions of all authors are not explicitly described here. Is it accepted in this journal style? (see, https://www.geoscientificmodel-development.net/submission.html#manuscriptcomposition)

**Reply: Thanks for bring up this point. We have revised the contribution part by explicitly stating contributions of all authors as follow (L824-827 in track-mode):**

"**YZ and MB defined the scope of the manuscript. YZ and KW designed the study and all the simulations. SY and DW developed the two-way coupled WRF-CMAQ code. KW conducted all the simulations and performed the analyses. KW prepared drafted the manuscript. YZ, SY, DW, JP, RM, JK, and MB reviewed and edited the manuscript.**"

**References:**

Glotfelty, T., J. He, and Y. Zhang (2017), Impact of future climate policy scenarios on air quality and aerosol-cloud interactions using an advanced version of CESM/CAM5: Part I. model evaluation for the current decadal simulations, Atmospheric Environment, 152, 222-239.

He, J., and Y. Zhang (2014), Improvement and further development in CESM/CAM5: Gasphase chemistry and inorganic aerosol treatments. Atmos. Chem. Phys. 14, 9171-9200. http://dx.doi.org/10.5194/acp-14-9171-2014.

Schaub, D., K. F. Boersma, J. W. Kaiser, A. K.Weiss, D. Folini, H. J. Eskes, and B. Buchmann (2006), Comparison of GOME tropospheric $NO_2$ columns with $NO_2$ profiles deduced from ground-based in situ measurements, Atmos. Chem. Phys., 6, 3211–3229.

---

## Author Comment (AC2) · 15 Jul 2021

**Responses to Reviewers**

**Anonymous Referee #2**

Wang et al. present a study which evaluates the performance of the of Two-way and Offline Coupled WRF v3.4 and CMAQ v5.0.2 over the Contiguous U.S for an extended time period (5 years). Previous works had experimental design deficiencies (e.g., differing physics, chemistry) that his work addresses. The importance of chemical meteorological
feedbacks are increasingly being recognized as essential for the prediction of both weather and atmospheric chemistry, and this work adds well to that body of work. Outside of a major comment w/ regards to the experimental design (cycling between 5-day periods), my main critique of the manuscript is the heavy reliance on the use of 5-year averages to discuss model performance and comparisons. This is also somewhat related to my major comment about cycling. I think it would benefit the community to examine and discuss seasonal spatial patterns (and thus reasons for model deficiencies), periods of peak aerosol and/or high ozone days (not just number of exceedance, but more details in how the model performs/evolves).

**Reply: We thank the reviewer for the general positive comments and recognizing the importance of this study. We have carefully revised our paper to fully address the reviewer's comments. Some major revisions have been performed including:**

**1) Old Figures 1-7 and Tables 1-2 are moved into supplementary materials. New Figures 1-13 and Tables 1-4 have been added by including the seasonal results.**
**2) A new figure (Figure 3) with the annual trend results has been added and corresponding analyses have been added in Sections 3.1.1, 3.2.1, and 3.2.2.**
**3) All texts in major Sections 3 and 4 have been updated with the seasonal results.**

**Please see our point-by-point detailed responses below.**

Major comment: Line 185: Are any fields cycled between consecutive 5-day simulations besides chemistry? (e.g., land surface fields?) I think this needs to be discussed in detail how it relates to the experiments. If they are reinitialized every 5 days, should the first day or two be considered in the comparisons? The deviation between the two simulations would likely increase as lead time increases. Here is really comparing 5 years of 5-day forecasts.

**Reply: The 5-day reinitialization are actually only applied to the meteorological fields (land use or land surface fields are assumed to be constant) and the chemistry fields are continuously simulated without any reinitialization. The same approach has been applied to both two-way WRF-CMAQ and WRF-only (providing the meteorological fields for offline CMAQ simulations) simulations. So any deviation for meteorology fields between two simulations are really more determined by the feedback processes. The reinitialization approach used in this work is very common practice for the air quality/chemistry transport models to ensure the accurate simulations of meteorological fields. The reinstallation may lead to some initial shock at very beginning of each 5-day, but wouldn't make significant impacts on simulation results based on our previous studies (see Wang et al., 2021).**

**We have added the following statements in revision (L190-196) to further clarify this issue.**

**"Two-way coupled WRF-CMAQ simulations are reinitialized every 5 days for meteorology fields only. We have conducted sensitivity simulations in the past (Wang et al., 2021) and found that a 5-day reinitialization frequency is more suitable to improve the overall simulation quality to make meteorology simulations as accurate as possible while preserving the two-way chemistry-meteorology feedbacks. The WRF-only simulations that are used to drive the offline CMAQ simulations apply the same reinitialization method to make sure any deviation between two simulations are more determined by the feedback processes"**

Minor comments: Sections 3.1.1 and 3.1.2: It would be much more helpful to at least break these comparisons up into summer vs. winter as some biases could be cancelling one another out.
**Reply: As suggested by the reviewer, we have broken up our previous 5-year annual average evaluation/analyses into summer vs. winter comparison for Sections 3 and 4. We have replaced most of previous figures and all tables in the original submission into seasonal comparison (i.e., summer vs winter). New analyses have also been added in the revision to reflect the changes of those figures and tables (see the track-mode revision on those changes).**

**In general, the new seasonal results show general consistent performance when comparing with old annual results (very limited cases of cancel-out of model biases in different seasons occur for example for T2, which has been explicitly pointed out in the revision; see L260-263 in the track-mode revision) and thus won't affect our previous conclusions based on the annual performance analyses. The seasonal results indeed shed more lights for some of our previous analyses and speculations. For example, the new seasonal T2 performance now can well support the O3 monthly and diurnal performance now. The speculation of model biases caused by biogenic emissions from BEIS on column HCHO in the original version has been eliminated after checking the season results (see L584-585 for changes in track-mode).**

Figure 4: The colors used in the top panel are very hard to distinguish.
**Reply: All the SWCF figures (now Figure 8 in revision) are updated with the new color scheme. Please note in both old and new plots, we intentionally use the cold-color only schemes to better represent SWCFs which only contain negative values.**

3.2.1. Annual average ozone is not really a useful diagnostic, I think showing summer only would be very beneficial.
**Reply: As suggested, we have replaced the annual average ozone with summer only and updated the corresponding texts in Section 3.2.1.**

3.2.2. Again, a seasonal analysis here would be more appropriate (i.e., winter is dominated by NO3, summer with OA (and SO4)).

**Reply: Both the spatial overlay plots for PM2.5 and scatter plots for PM compositions have been broken up into summer vs winter comparison and the corresponding texts are updated in Section 3.2.2.**

Figure 5a-b: You could shift the color limits by 20 ppb.
**Reply: Both figures (now Figure 9a-b in revision) have been replaced by summer ones and the scales have been adjusted to from 30 to 60 ppb as suggested.**

Figure 8. Why not just use more colors instead of the varying dot sizes – hard to distinguish.
**Reply: The figure (now Figure 14 in revision) has been updated by using different colors instead of varying dot sizes.**

Figure 10. Looks to be some weird striping for O3.
**Reply: The striping seems to be caused by the WRF-CMAQ interface to deal with feedback interactions among multiple CPUs while conducting parallel computing, which hardly can affect any conclusions.**

**Reference:**

**Wang, K., Y. Zhang, and K. Yahya (2021), Decadal application of WRF/Chem over the continental U.S.: Simulation design, sensitivity simulations, and climatological model evaluation, Atmospheric Environment, 118331.**

---

## Author Response (AR2)

**Responses to the Reviewer**

General Comments:
The revised manuscript 'A comprehensive study of two-way and offline coupled WRF v3.4 and CMAQ v5.0.2 over the contiguous U.S.: Performance evaluation and impacts of chemistry-meteorology feedbacks on air quality' presented the comprehensive comparison of offline (i.e., traditional) CMAQ and two-way coupled CMAQ over the CONUS. I appreciate the authors' work to address my concerns, especially by adding the trend analysis. As commented by other reviewer, seasonal analysis has been added and figures are replaced appropriately in the supplemental material. The presentation quality has been significantly improved from original version. Here I have two further comments on the revised manuscript. Please clarify these issues.

**Reply: We thank the reviewer for carefully reviewing our revision. We have further revised our paper to address the reviewer's additional comments. Please see our point-by-point responses below.**

1. Long-term trends of meteorology and pollutants: I appreciate to include 5-yr trends analysis in the revised manuscript. The additional discussion in Fig. 3 is important to consider the modeling performances. In the discussion of trends in PM2.5, the authors stated that "Overall, the model performs well for PM2.5 for most of years and better over CSN than IMPROVE sites with general underpredictions in most years. The observations for both CSN and IMPROVE show a general decreasing trend (except for 2010 over CSN) especially over IMPROVE sites.". I impressed different understanding in this trend found over CSN. I guess CSN also showed gradual decreasing trend from 2008 to 2012 but the year of 2009 posed strong drop. This lower value found in 2009 is consistently seen both by model and observation, but interestingly, IMPROVE sites did not the drop on 2009. If there is specific reason to cause this drop, it is useful to include the short statement on this feature.
**Reply: The reviewer is right that CSN indeed shows gradual decreasing trend except for 2009 instead of 2010 as we initially thought in the paper. According to EPA (2012), the strong drop of PM$_{2.5}$ in 2009 is due to a few reasons including many national and local regulations that are imposed, the contribution of economic slowdown to cleaner air conditions and also favorable meteorological conditions to lower air pollution levels in 2009. The impacts are more apparent over CSN sites mainly composed of urban/suburban areas than IMPROVE sites mainly composed of remote areas and national parks. The above points have been added in the revision lines 498-504 (in the track-mode file).**

2. Satellite data comparison for column abundant: I understood that AK is not considered and only the valid pixels by satellite observations are considered when paring model results. However, for example as shown in Fig. 12, satellite observed NO2 and HCHO showed deficit at northern border of domain (over Canada) whereas model calculated values over this area. If the deficit is treated as same in model analysis, the deficit (marked by white color in this figure) grid should be consistent (this concern is also noticed on AOD presented in Fig. 4 and CDNC presented in Figs. 6 and 7). It is highly recommended to confirm the analysis procedure again. Regarding this figure, winter time HCHO posed inhomogeneous signals. In the revised manuscript, "except for column HCHO in winter" is stated in line 553 (track-mode), but is the satellite measurement itself reliable?

**Reply: To further address the reviewer's concern on the satellite and simulation data pairing and comparison, we have updated all the plots that containing missing satellite data (considered as deficit data) by excluding those data from simulation plots as well. The domain mean values on the plots have been updated as well. The updated plots include AOD in Fig. 4, CDNC in Figs. 6 and 7, and $NO_2$ and HCHO in Fig. 12. Now both updated satellite and simulation plots contain the consistent data for comparison.**

**As also indicated by the reviewer, the satellite measurements on HCHO may indeed have higher uncertainties in winter than summer. According to Stavrakou et al. (2009), the air mass factors used for HCHO column calculation may bear ~18% error under clear sky conditions to ~50% error for very cloudy conditions. The winter typically has higher cloud cover than summer (See Figs. 6 and 7) and thus higher uncertainties for HCHO column. This point has been added in the revision lines 567-570 (in the track-mode file).**

Technical points:
Fig. 2: It is better to add DJF and JJA for the legend of TMPA and PRISM.
**Reply: The DJF and JJA have been added.**

Fig. 3: The title of y-axis shown as "RH" should be "RH2" to be consistent in the discussion in main text.
**Reply: It's been fixed.**

**References:**

Stavrakou, T., Müller, J.-F., De Smedt, I., Van Roozendael, M., van der Werf, G. R., Giglio, L., and Guenther, A.: Global emissions of non-methane hydrocarbons deduced from SCIAMACHY formaldehyde columns through 2003–2006, Atmos. Chem. Phys., 9, 3663–3679, doi:10.5194/acp-9-3663-2009, 2009.

U.S. EPA: Our nation's air status and trends through 2010, EPA-454/R-12-001, February 2012, https://www.epa.gov/sites/default/files/2017-11/documents/trends_brochure_2010.pdf, 2012.